# Adversarial Examples are not Bugs, they are Features

**Andrew Ilyas**[*]
MIT
ailyas@mit.edu

**Shibani Santurkar**[*]
MIT
shibani@mit.edu

**Dimitris Tsipras**[*]
MIT
tsipras@mit.edu

**Logan Engstrom**[*]
MIT
engstrom@mit.edu

**Brandon Tran**
MIT
btran115@mit.edu

**Aleksander Mądry**
MIT
madry@mit.edu

## Abstract

Adversarial examples have attracted significant attention in machine learning, but the reasons for their existence and pervasiveness remain unclear. We demonstrate that adversarial examples can be directly attributed to the presence of *non-robust features*: features (derived from patterns in the data distribution) that are highly predictive, yet brittle and (thus) incomprehensible to humans. After capturing these features within a theoretical framework, we establish their widespread existence in standard datasets. Finally, we present a simple setting where we can rigorously tie the phenomena we observe in practice to a *misalignment* between the (human-specified) notion of robustness and the inherent geometry of the data.

## 1 Introduction

The pervasive brittleness of deep neural networks [Sze+14; Eng+19b; HD19; Ath+18] has attracted significant attention in recent years. Particularly worrisome is the phenomenon of *adversarial examples* [Big+13; Sze+14], imperceptibly perturbed natural inputs that induce erroneous predictions in state-of-the-art classifiers. Previous work has proposed a variety of explanations for this phenomenon, ranging from theoretical models [Sch+18; BPR18] to arguments based on concentration of measure in high-dimensions [Gil+18; MDM18; Sha+19a]. These theories, however, are often unable to fully capture behaviors we observe in practice (we discuss this further in Section 5).

More broadly, previous work in the field tends to view adversarial examples as aberrations arising either from the high dimensional nature of the input space or statistical fluctuations in the training data [GSS15; Gil+18]. From this point of view, it is natural to treat adversarial robustness as a goal that can be disentangled and pursued independently from maximizing accuracy [Mad+18; SHS19; Sug+19], either through improved standard regularization methods [TG16] or pre/post-processing of network inputs/outputs [Ues+18; CW17a; He+17].

In this work, we propose a new perspective on the phenomenon of adversarial examples. In contrast to the previous models, we cast adversarial vulnerability as a fundamental consequence of the dominant supervised learning paradigm. Specifically, we claim that:

*Adversarial vulnerability is a direct result of sensitivity to well-generalizing features in the data.*

Recall that we usually train classifiers to *solely* maximize (distributional) accuracy. Consequently, classifiers tend to use *any* available signal to do so, even those that look incomprehensible to humans. After all, the presence of "a tail" or "ears" is no more natural to a classifier than any other equally predictive feature. In fact, we find that standard ML datasets *do* admit highly predictive

yet imperceptible features. We posit that our models learn to rely on these "non-robust" features, leading to adversarial perturbations that exploit this dependence.[2]

Our hypothesis also suggests an explanation for *adversarial transferability*: the phenomenon that perturbations computed for one model often transfer to other, independently trained models. Since any two models are likely to learn similar non-robust features, perturbations that manipulate such features will apply to both. Finally, this perspective establishes adversarial vulnerability as a human-centric phenomenon, since, from the standard supervised learning point of view, non-robust features can be as important as robust ones. It also suggests that approaches aiming to enhance the interpretability of a given model by enforcing "priors" for its explanation [MV15; OMS17; Smi+17] actually hide features that are *"meaningful"* and *predictive* to standard models. As such, producing *human*-meaningful explanations that remain faithful to underlying models cannot be pursued independently from the training of the models themselves.

To corroborate our theory, we show that it is possible to disentangle robust from non-robust features in standard image classification datasets. Specifically, given a training dataset, we construct:

1. **A "robustified" version for robust classification (Figure 1a)**[3] **.** We are able to effectively remove non-robust features from a dataset. Concretely, we create a training set (semantically similar to the original) on which *standard training* yields *good robust accuracy* on the *original, unmodified* test set. This finding establishes that adversarial vulnerability is not necessarily tied to the standard training framework, but is also a property of the dataset.

2. **A "non-robust" version for standard classification (Figure 1b)**[2]**.** We are also able to construct a training dataset for which the inputs are nearly identical to the originals, but all appear incorrectly labeled. In fact, the inputs in the new training set are associated to their labels only through *small adversarial perturbations* (and hence utilize only non-robust features). Despite the lack of any predictive human-visible information, training on this dataset yields good accuracy on the *original, unmodified* test set. This demonstrates that adversarial perturbations can arise from flipping features in the data that are useful for classification of correct inputs (hence not being purely aberrations).

Finally, we present a concrete classification task where the connection between adversarial examples and non-robust features can be studied rigorously. This task consists of separating Gaussian distributions, and is loosely based on the model presented in Tsipras et al. [Tsi+19], while expanding upon it in a few ways. First, adversarial vulnerability in our setting can be precisely quantified as a difference between the intrinsic data geometry and that of the adversary's perturbation set. Second, robust training yields a classifier which utilizes a geometry corresponding to a combination of these two. Lastly, the gradients of standard models can be significantly misaligned with the inter-class direction, capturing a phenomenon that has been observed in practice [Tsi+19].

## 2   The Robust Features Model

We begin by developing a framework, loosely based on the setting of Tsipras et al. [Tsi+19], that enables us to rigorously refer to "robust" and "non-robust" features. In particular, we present a set of definitions which allow us to formally describe our setup, theoretical results, and empirical evidence.

**Setup.**   We study binary classification, where input-label pairs $(x, y) \in \mathcal{X} \times \{\pm 1\}$ are sampled from a distribution $\mathcal{D}$; the goal is to learn a classifier $C : \mathcal{X} \to \{\pm 1\}$ predicting $y$ given $x$.

We define a *feature* to be a function mapping from the input space $\mathcal{X}$ to real numbers, with the set of all features thus being $\mathcal{F} = \{f : \mathcal{X} \to \mathbb{R}\}$. For convenience, we assume that the features in $\mathcal{F}$ are shifted/scaled to be mean-zero and unit-variance (i.e., so that $\mathbb{E}_{(x,y)\sim\mathcal{D}}[f(x)] = 0$ and $\mathbb{E}_{(x,y)\sim\mathcal{D}}[f(x)^2] = 1$), making the following definitions scale-invariant. Note that this definition captures what we abstractly think of as features (e.g., a function capturing how "furry" an image is).

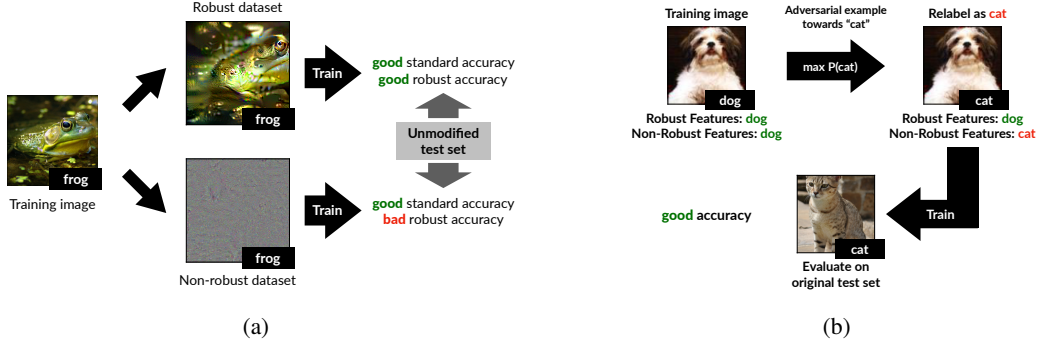

(a)

(b)

Figure 1: A conceptual diagram of the experiments of Section 3: (a) we disentangle features into robust and non-robust (Section 3.1), (b) we construct a dataset which appears mislabeled to humans (via adversarial examples) but results in good accuracy on the original test set (Section 3.2).

**Useful, robust, and non-robust features.**   We now define the key concepts required for formulating our framework. To this end, we categorize features in the following manner:

- $\rho$**-useful features**: For a given distribution $\mathcal{D}$, we call a feature $f$ $\rho$-*useful* ($\rho > 0$) if it is correlated with the true label in expectation, that is if

$$\mathbb{E}_{(x,y)\sim\mathcal{D}}[y \cdot f(x)] \geq \rho. \tag{1}$$

  We then define $\rho_{\mathcal{D}}(f)$ as the largest $\rho$ for which feature $f$ is $\rho$-useful under distribution $\mathcal{D}$. (Note that if a feature $f$ is negatively correlated with the label, then $-f$ is useful instead.) Crucially, a linear classifier trained on $\rho$-useful features can attain non-trivial performance.

- $\gamma$**-robustly useful features**: Suppose we have a $\rho$-useful feature $f$ ($\rho_{\mathcal{D}}(f) > 0$). We refer to $f$ as a *robust feature* (formally a $\gamma$-robustly useful feature for $\gamma > 0$) if, under adversarial perturbation (for some specified set of valid perturbations $\Delta$), $f$ remains $\gamma$-useful. Formally, if we have that

$$\mathbb{E}_{(x,y)\sim\mathcal{D}}\left[\inf_{\delta\in\Delta(x)} y \cdot f(x+\delta)\right] \geq \gamma. \tag{2}$$

- **Useful, non-robust features**: A *useful, non-robust feature* is a feature which is $\rho$-useful for some $\rho$ bounded away from zero, but is not a $\gamma$-robust feature for any $\gamma \geq 0$. These features help with classification in the standard setting, but may hinder accuracy in the adversarial setting, as the correlation with the label can be flipped.

**Classification.**   In our framework, a classifier $C = (F, w, b)$ is comprised of a set of features $F \subseteq \mathcal{F}$, a weight vector $w$, and a scalar bias $b$. For an input $x$, the classifier predicts the label $y$ as

$$C(x) = \text{sgn}\left(b + \sum_{f\in F} w_f \cdot f(x)\right).$$

For convenience, we denote the set of features learned by a classifier $C$ as $F_C$.

**Standard Training.**   Training a classifier is performed by minimizing a loss function $\mathcal{L}_{\theta}(x, y)$ over input-label pairs $(x, y)$ from the training set (via *empirical risk minimization* (ERM)) that decreases with the correlation between the weighted combination of the features and the label. When minimizing classification loss, *no distinction* exists between robust and non-robust features: the only distinguishing factor of a feature is its $\rho$-usefulness. Furthermore, the classifier will utilize *any* $\rho$-useful feature in $F$ to decrease the loss of the classifier.

**Robust training.**   In the presence of an *adversary*, any useful but non-robust features can be made *anti-correlated* with the true label, leading to adversarial vulnerability. Therefore, ERM is no longer sufficient to train classifiers that are robust, and we need to explicitly account for the effect of the

adversary on the classifier. To do so, we use an *adversarial* loss function that can discern between robust and non-robust features [Mad+18]:

$$\mathbb{E}_{(x,y)\sim\mathcal{D}}\left[\max_{\delta\in\Delta(x)}\mathcal{L}_\theta(x+\delta,y)\right],\tag{3}$$

for an appropriately defined set of perturbations $\Delta$. Since the adversary can exploit non-robust features to degrade classification accuracy, minimizing this adversarial loss [GSS15; Mad+18] can be viewed as explicitly preventing the classifier from relying on non-robust features.

**Remark.**    We want to note that even though this framework enables us to describe and predict the outcome of our experiments, it does not capture the notion of non-robust features exactly as we intuitively might think of them. For instance, in principle, our theoretical framework would allow for useful non-robust features to arise as combinations of useful robust features and useless non-robust features [Goh19b]. These types of constructions, however, are actually precluded by our experimental results (for instance, the classifiers trained in Section 3 would not generalize). This shows that our experimental findings capture a stronger, more fine-grained statement than our formal definitions are able to express. We view bridging this gap as an interesting direction for future work.

## 3    Finding Robust (and Non-Robust) Features

The central premise of our proposed framework is that there exist both robust and non-robust features that constitute useful signals for standard classification. We now provide evidence in support of this hypothesis by disentangling these two sets of features (see conceptual description in Figure 1).

On one hand, we will construct a "robustified" dataset, consisting of samples that primarily contain robust features. Using such a dataset, we are able to train robust classifiers (with respect to the standard test set) using standard (i.e., non-robust) training. This demonstrates that robustness can arise by *removing* certain features from the dataset (as, overall, the new dataset contains less information about the original training set). Moreover, it provides evidence that adversarial vulnerability is caused by non-robust features and is not inherently tied to the standard training framework.

On the other hand, we will construct datasets where the input-label association is based purely on non-robust features (and thus the resulting dataset appears *completely* mislabeled to humans). We show that this dataset suffices to train a classifier with good performance on the standard test set. This indicates that natural models use *non-robust features* to make predictions, even in the presence of robust features. These features *alone* are sufficient for non-trivial generalization to *natural images*, indicating that they are indeed predictive, rather than artifacts of finite-sample overfitting.

### 3.1    Disentangling robust and non-robust features

Recall that the features a classifier learns to rely on are based purely on how useful these features are for (standard) generalization. Thus, under our conceptual framework, if we can ensure that only robust features are useful, standard training should result in a robust classifier. Unfortunately, we cannot directly manipulate the features of very complex, high-dimensional datasets. Instead, we will leverage a robust model and modify our dataset to contain only the features relevant to that model.

Conceptually, given a *robust* (i.e., adversarially trained [Mad+18]) model $C$, we aim to construct a distribution $\widehat{\mathcal{D}}_R$ for which features used by $C$ are as useful as they were on the original distribution $\mathcal{D}$ while ensuring that the rest of the features are not useful. In terms of our formal framework:

$$\mathbb{E}_{(x,y)\sim\widehat{\mathcal{D}}_R}\left[f(x)\cdot y\right]=\begin{cases}\mathbb{E}_{(x,y)\sim\mathcal{D}}\left[f(x)\cdot y\right] & \text{if } f\in F_C\\0 & \text{otherwise,}\end{cases}\tag{4}$$

where $F_C$ again represents the set of features utilized by $C$.

We will construct a training set for $\widehat{\mathcal{D}}_R$ via a one-to-one mapping $x\mapsto x_r$ from the original training set for $\mathcal{D}$. In the case of a deep neural network, $F_C$ corresponds to exactly the set of activations in the penultimate layer (since these correspond to inputs to a linear classifier). To ensure that features used by the model are equally useful under both training sets, we (approximately) enforce all features in $F_C$ to have similar values for both $x$ and $x_r$ through the following optimization:

$$\min_{x_r}\|g(x_r)-g(x)\|_2,\tag{5}$$

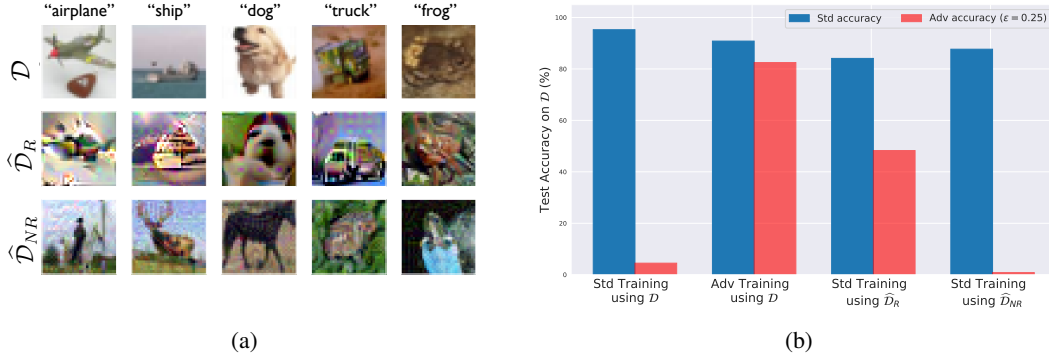

(a)                                               (b)

Figure 2: **(a)**: Random samples from our variants of the CIFAR-10 [Kri09] training set: the original training set; the *robust training set* $\widehat{\mathcal{D}}_R$, restricted to features used by a robust model; and the *non-robust training set* $\widehat{\mathcal{D}}_{NR}$, restricted to features relevant to a standard model (labels appear incorrect to humans). **(b)**: Standard and robust accuracy on the CIFAR-10 test set ($\mathcal{D}$) for models trained with: (i) standard training (on $\mathcal{D}$) ; (ii) standard training on $\widehat{\mathcal{D}}_{NR}$; (iii) adversarial training (on $\mathcal{D}$); and (iv) standard training on $\widehat{\mathcal{D}}_R$. Models trained on $\widehat{\mathcal{D}}_R$ and $\widehat{\mathcal{D}}_{NR}$ reflect the original models used to create them: notably, standard training on $\widehat{\mathcal{D}}_R$ yields nontrivial robust accuracy. Results for Restricted-ImageNet [Tsi+19] are in D.8 Figure 12.

where $x$ is the original input and $g$ is the mapping from $x$ to the representation layer. We optimize this objective using (normalized) gradient descent (see details in Appendix C).

Since we don't have access to features outside $F_C$, there is no way to ensure that the expectation in (4) is zero for all $f \notin F_C$. To approximate this condition, we choose the starting point of gradient descent for the optimization in (5) to be an input $x_0$ which is drawn from $\mathcal{D}$ independently of the label of $x$ (we also explore sampling $x_0$ from noise in Appendix D.1). This choice ensures that any feature present in that input will not be useful since they are not correlated with the label in expectation over $x_0$. The underlying assumption here is that, when performing the optimization in (5), features that are not being directly optimized (i.e., features outside $F_C$) are not affected. We provide pseudocode for the construction in Figure 5 (Appendix C).

Given the new training set for $\widehat{\mathcal{D}}_R$ (a few random samples are visualized in Figure 2a), we train a classifier using standard (non-robust) training. We then test this classifier on the original test set (i.e. $\mathcal{D}$). The results (Figure 2b) indicate that the classifier learned using the new dataset attains good accuracy in *both standard and adversarial settings* (see additional evaluation in Appendix D.2.) [4].

As a control, we repeat this methodology using a standard (non-robust) model for $C$ in our construction of the dataset. Sample images from the resulting "non-robust dataset" $\widehat{\mathcal{D}}_{NR}$ are shown in Figure 2a—they tend to resemble more the source image of the optimization $x_0$ than the target image $x$. We find that training on this dataset leads to good standard accuracy, yet yields almost no robustness (Figure 2b). We also verify that this procedure is not simply a matter of encoding the weights of the original model—we get the same results for both $\widehat{\mathcal{D}}_R$ and $\widehat{\mathcal{D}}_{NR}$ if we train with different architectures than that of the original models.

Overall, our findings corroborate the hypothesis that adversarial examples can arise from (non-robust) features of the data itself. By filtering out non-robust features from the dataset (e.g. by restricting the set of available features to those used by a robust model), one can train a significantly more robust model using *standard training*.

## 3.2 Non-robust features suffice for standard classification

The results of the previous section show that by restricting the dataset to only contain features that are used by a robust model, standard training results in classifiers that are significantly more robust.

This suggests that when training on the standard dataset, non-robust features take on a large role in the resulting learned classifier. Here we will show that this is not merely incidental. In particular, we demonstrate that non-robust features *alone* suffice for standard generalization— i.e., a model trained solely on non-robust features can generalize to the *standard* test set.

To show this, we construct a dataset where the only features that are useful for classification are *non-robust* features (or in terms of our formal model from Section 2, all features $f$ that are $\rho$-useful are non-robust). To accomplish this, we modify each input-label pair $(x, y)$ as follows. We select a target class $t$ either (a) uniformly at random (hence features become uncorrelated with the labels) or (b) deterministically according to the source class (e.g. permuting the labels). Then, we add a small adversarial perturbation to $x$ to cause it to be classified as $t$ by a standard model:

$$x_{adv} = \underset{\|x'-x\| \leq \varepsilon}{\arg\min} \ L_C(x', t), \tag{6}$$

where $L_C$ is the loss under a standard (non-robust) classifier $C$ and $\varepsilon$ is a small constant. The resulting inputs are indistinguishable from the originals (Appendix D Figure 9)—to a human observer, it thus appears that the label $t$ assigned to the modified input is simply incorrect. The resulting input-label pairs $(x_{adv}, t)$ make up the new training set (pseudocode in Appendix C Figure 6).

Now, since $\|x_{adv} - x\|$ is small, by definition the robust features of $x_{adv}$ are still correlated with class $y$ (and not $t$) in expectation over the dataset. After all, humans still recognize the original class. On the other hand, since every $x_{adv}$ is strongly classified as $t$ by a standard classifier, it must be that some of the non-robust features are now strongly correlated with $t$ (in expectation).

In the case where $t$ is chosen at random, the robust features are originally uncorrelated with the label $t$ (in expectation), and after the perturbation can be only slightly correlated (hence being significantly less useful for classification than before) [5]. Formally, we aim to construct a dataset $\widehat{\mathcal{D}}_{rand}$ where

$$\mathbb{E}_{(x,y)\sim\widehat{\mathcal{D}}_{rand}} [y \cdot f(x)] \begin{cases} > 0 & \text{if } f \text{ non-robustly useful under } \mathcal{D}, \\ \simeq 0 & \text{otherwise.} \end{cases} \tag{7}$$

In contrast, when $t$ is chosen deterministically based on $y$, the robust features actually point *away* from the assigned label $t$. In particular, all of the inputs labeled with class $t$ exhibit *non-robust features* correlated with $t$, but robust features correlated with the original class $y$. Thus, robust features on the original training set provide significant predictive power on the training set, but will actually hurt generalization on the standard test set. Formally, our goal is to construct $\widehat{\mathcal{D}}_{det}$ such that

$$\mathbb{E}_{(x,y)\sim\widehat{\mathcal{D}}_{det}} [y \cdot f(x)] \begin{cases} > 0 & \text{if } f \text{ non-robustly useful under } \mathcal{D}, \\ < 0 & \text{if } f \text{ robustly useful under } \mathcal{D} \\ \in \mathbb{R} & \text{otherwise } (f \text{ not useful under } \mathcal{D}). \end{cases} \tag{8}$$

We find that standard training on these datasets actually generalizes to the *original* test set, as shown in Table 1). This indicates that non-robust features are indeed useful for classification in the standard setting. Remarkably, even training on $\widehat{\mathcal{D}}_{det}$ (where all the robust features are correlated with the wrong class), results in a well-generalizing classifier. This indicates that non-robust features can be picked up by models during standard training, even in the presence of predictive *robust features* [6]

### 3.3 Transferability can arise from non-robust features

One of the most intriguing properties of adversarial examples is that they *transfer* across models with different architectures and independently sampled training sets [Sze+14; PMG16; CRP19]. Here, we show that this phenomenon can in fact be viewed as a natural consequence of the existence of non-robust features. Recall that, according to our main thesis, adversarial examples can arise as a result of perturbing well-generalizing, yet brittle features. Given that such features are inherent to the data distribution, different classifiers trained on independent samples from that distribution are likely to utilize similar non-robust features. Consequently, perturbations constructed by exploiting non-robust features learned by one classifier will transfer to other classifiers utilizing similar features.

In order to illustrate and corroborate this hypothesis, we train five different architectures on the dataset generated in Section 3.2 (adversarial examples with deterministic labels) for a standard ResNet-50 [He+16]. Our hypothesis would suggest that architectures which learn better from this training set (in terms of performance on the standard test set) are more likely to learn similar non-robust features to the original classifier. Indeed, we find that the test accuracy of each architecture is predictive of how often adversarial examples transfer from the original model to standard classifiers with that architecture (Figure 3). In a similar vein, Nakkiran [Nak19a] constructs a set of adversarial perturbations that is explicitly non-transferable and finds that these perturbations cannot be used to learn a good classifier. These findings thus corroborate our hypothesis that adversarial transferability arises when models learn similar brittle features of the underlying dataset.

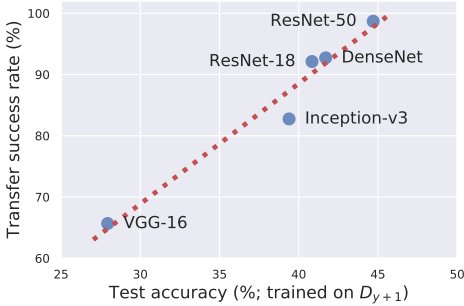

Figure 3: Transfer rate of adversarial examples from a ResNet-50 to different architectures alongside test set performance of these architecture when trained on the dataset generated in Section 3.2. Architectures more susceptible to transfer attacks also performed better on the standard test set supporting our hypothesis that adversarial transferability arises from using similar *non-robust features*.

| Source Dataset | Dataset | |
| --- | --- | --- |
| | CIFAR-10 | ImageNet$_R$ |
| $\mathcal{D}$ | 95.3% | 96.6% |
| $\widehat{\mathcal{D}}_{rand}$ | 63.3% | 87.9% |
| $\widehat{\mathcal{D}}_{det}$ | 43.7% | 64.4% |

Table 1: Test accuracy (on $\mathcal{D}$) of classifiers trained on the $\mathcal{D}$, $\widehat{\mathcal{D}}_{rand}$, and $\widehat{\mathcal{D}}_{det}$ training sets created using a standard (non-robust) model. For both $\widehat{\mathcal{D}}_{rand}$ and $\widehat{\mathcal{D}}_{det}$, only non-robust features correspond to useful features on both the train set and $\mathcal{D}$. These datasets are constructed using adversarial perturbations of $x$ towards a class $t$ (random for $\widehat{\mathcal{D}}_{rand}$ and deterministic for $\widehat{\mathcal{D}}_{det}$); the resulting images are relabeled as $t$.

## 4 A Theoretical Framework for Studying (Non)-Robust Features

The experiments from the previous section demonstrate that the conceptual framework of robust and non-robust features is strongly predictive of the empirical behavior of state-of-the-art models on real-world datasets. In order to further strengthen our understanding of the phenomenon, we instantiate the framework in a concrete setting that allows us to theoretically study various properties of the corresponding model. Our model is similar to that of Tsipras et al. [Tsi+19] in the sense that it contains a dichotomy between robust and non-robust features, extending upon it in a few ways: a) the adversarial vulnerability can be explicitly expressed as a difference between the inherent data metric and the $\ell_2$ metric, b) robust learning corresponds exactly to learning a combination of these two metrics, c) the gradients of robust models align better with the adversary's metric.

**Setup.** We study a simple problem of *maximum likelihood classification* between two Gaussian distributions. In particular, given samples $(x, y)$ sampled from $\mathcal{D}$ according to

$$y \overset{\text{u.a.r.}}{\sim} \{-1, +1\}, \qquad x \sim \mathcal{N}(y \cdot \boldsymbol{\mu}_*, \boldsymbol{\Sigma}_*), \tag{9}$$

our goal is to learn parameters $\Theta = (\boldsymbol{\mu}, \boldsymbol{\Sigma})$ such that

$$\Theta = \arg\min_{\boldsymbol{\mu}, \boldsymbol{\Sigma}} \mathbb{E}_{(x,y) \sim \mathcal{D}} \left[ \ell(x; y \cdot \boldsymbol{\mu}, \boldsymbol{\Sigma}) \right], \tag{10}$$

where $\ell(x; \boldsymbol{\mu}, \boldsymbol{\Sigma})$ represents the Gaussian negative log-likelihood (NLL) function. Intuitively, we find the parameters $\boldsymbol{\mu}, \boldsymbol{\Sigma}$ which maximize the likelihood of the sampled data under the given model. Classification can be accomplished via likelihood test: given an unlabeled sample $x$, we predict $y$ as

$$y = \arg\max_{y} \ell(x; y \cdot \boldsymbol{\mu}, \boldsymbol{\Sigma}) = \text{sign}\left( x^\top \boldsymbol{\Sigma}^{-1} \boldsymbol{\mu} \right).$$

In turn, the *robust analogue* of this problem arises from replacing $\ell(x; y \cdot \boldsymbol{\mu}, \boldsymbol{\Sigma})$ with the NLL under adversarial perturbation. The resulting robust parameters $\Theta_r$ can be written as

$$\Theta_r = \arg\min_{\boldsymbol{\mu}, \boldsymbol{\Sigma}} \mathbb{E}_{(x,y) \sim \mathcal{D}} \left[ \max_{\|\delta\|_2 \leq \varepsilon} \ell(x + \delta; y \cdot \boldsymbol{\mu}, \boldsymbol{\Sigma}) \right], \qquad (11)$$

A detailed analysis appears in Appendix E—here we present a high-level overview of the results.

**(1) Vulnerability from metric misalignment (non-robust features).** Note that in this model, one can rigorously refer to an *inner product* (and thus a metric) induced by the features. In particular, one can view the learned parameters of a Gaussian $\Theta = (\boldsymbol{\mu}, \boldsymbol{\Sigma})$ as defining an inner product over the input space given by $\langle x, y \rangle_\Theta = (x - \boldsymbol{\mu})^\top \boldsymbol{\Sigma}^{-1} (y - \boldsymbol{\mu})$. This in turn induces the Mahalanobis distance, which represents how a change in the input affects the features of the classifier. This metric is not necessarily aligned with the metric in which the adversary is constrained, the $\ell_2$-norm. Actually, we show that adversarial vulnerability arises exactly as a *misalignment* of these two metrics.

**Theorem 1** (Adversarial vulnerability from misalignment). *Consider an adversary whose perturbation is determined by the "Lagrangian penalty" form of (11), i.e.*

$$\max_\delta \ell(x + \delta; y \cdot \boldsymbol{\mu}, \boldsymbol{\Sigma}) - C \cdot \|\delta\|_2,$$

*where $C \geq \frac{1}{\sigma_{min}(\boldsymbol{\Sigma}_*)}$ is a constant trading off NLL minimization and the adversarial constraint (the bound on $C$ ensures the problem is concave). Then, the adversarial loss $\mathcal{L}_{adv}$ incurred by $(\boldsymbol{\mu}, \boldsymbol{\Sigma})$ is*

$$\mathcal{L}_{adv}(\Theta) - \mathcal{L}(\Theta) = tr \left[ \left( I + (C \cdot \boldsymbol{\Sigma}_* - I)^{-1} \right)^2 \right] - d,$$

*and, for a fixed $tr(\boldsymbol{\Sigma}_*) = k$ the above is minimized by $\boldsymbol{\Sigma}_* = \frac{k}{d} \boldsymbol{I}$.*

In fact, note that such a misalignment corresponds precisely to the existence of *non-robust features*—"small" changes in the adversary's metric along certain directions can cause large changes under the notion of distance established by the parameters (illustrated in Figure 4).

**(2) Robust Learning.** The (non-robust) maximum likelihood estimate is $\Theta = \Theta^*$, and thus the vulnerability for the standard MLE depends entirely on the data distribution. The following theorem characterizes the behaviour of the learned parameters in the robust problem (we study a slight relaxation of (11) that becomes exact exponentially fast as $d \to \infty$, see Appendix E.3.3). In fact, we can prove (Section E.3.4) that performing (sub)gradient descent on the inner maximization (known as *adversarial training* [GSS15; Mad+18]) yields exactly $\Theta_r$. We find that as the perturbation budget $\varepsilon$ increases, the metric induced by the classifier *mixes* $\ell_2$ and the metric induced by the data features.

**Theorem 2** (Robustly Learned Parameters). *Just as in the non-robust case, $\boldsymbol{\mu}_r = \boldsymbol{\mu}^*$, i.e. the true mean is learned. For the robust covariance $\boldsymbol{\Sigma}_r$, there exists an $\varepsilon_0 > 0$, such that for any $\varepsilon \in [0, \varepsilon_0)$,*

$$\boldsymbol{\Sigma}_r = \frac{1}{2} \boldsymbol{\Sigma}_* + \frac{1}{\lambda} \cdot \boldsymbol{I} + \sqrt{\frac{1}{\lambda} \cdot \boldsymbol{\Sigma}_* + \frac{1}{4} \boldsymbol{\Sigma}_*^2}, \qquad where \qquad \Omega\left( \frac{1 + \varepsilon^{1/2}}{\varepsilon^{1/2} + \varepsilon^{3/2}} \right) \leq \lambda \leq O\left( \frac{1 + \varepsilon^{1/2}}{\varepsilon^{1/2}} \right).$$

The effect of robust optimization under an $\ell_2$-constrained adversary is visualized in Figure 4. As $\epsilon$ grows, the learned covariance becomes more aligned with identity. For instance, we can see that the classifier learns to be less sensitive in certain directions, despite their usefulness for classification.

**(3) Gradient Interpretability.** Tsipras et al. [Tsi+19] observe that gradients of robust models tend to look more semantically meaningful. It turns out that under our model, this behaviour arises as a natural consequence of Theorem 2. In particular, we show that the resulting robustly learned parameters cause the gradient of the linear classifier and the vector connecting the means of the two distributions to better align (in a worst-case sense) under the $\ell_2$ inner product.

**Theorem 3** (Gradient alignment). *Let $f(x)$ and $f_r(x)$ be monotonic classifiers based on the linear separator induced by standard and $\ell_2$-robust maximum likelihood classification, respectively. The maximum angle formed between the gradient of the classifier (wrt input) and the vector connecting the classes can be smaller for the robust model:*

$$\min_{\boldsymbol{\mu}} \frac{\langle \boldsymbol{\mu}, \nabla_x f_r(x) \rangle}{\|\boldsymbol{\mu}\| \cdot \|\nabla_x f_r(x)\|} > \min_{\boldsymbol{\mu}} \frac{\langle \boldsymbol{\mu}, \nabla_x f(x) \rangle}{\|\boldsymbol{\mu}\| \cdot \|\nabla_x f(x)\|}.$$

Figure 4 illustrates this phenomenon in the two-dimensional case, where $\ell_2$-robustness causes the gradient direction to become increasingly aligned with the vector between the means ($\boldsymbol{\mu}$).

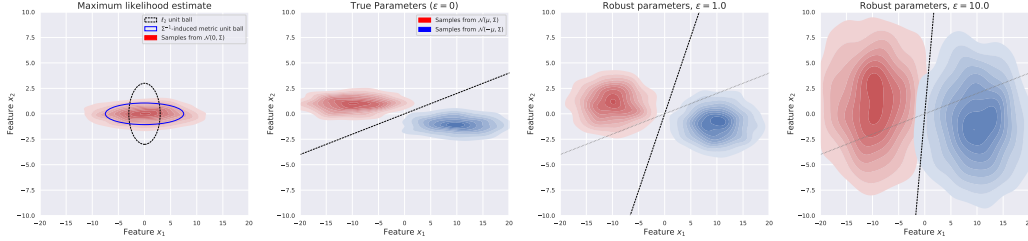

Figure 4: An empirical demonstration of the effect illustrated by Theorem 2—as the adversarial perturbation budget $\varepsilon$ is increased, the learned mean $\boldsymbol{\mu}$ remains constant, but the learned covariance "blends" with the identity matrix, effectively adding uncertainty onto the non-robust feature.

**Discussion.** Our analysis suggests that rather than offering quantitative classification benefits, a natural way to view the role of robust optimization is as enforcing a *prior* over the features learned by the classifier. In particular, training with an $\ell_2$-bounded adversary prevents the classifier from relying heavily on features which induce a metric dissimilar to the $\ell_2$ metric. The strength of the adversary then allows for a trade-off between the enforced prior, and the data-dependent features.

**Robustness and accuracy.** Note that in the setting described so far, robustness *can* be at odds with accuracy since robust training prevents us from learning the most accurate classifier (a similar conclusion is drawn in [Tsi+19]). However, we note that there are very similar settings where non-robust features manifest themselves in the same way, yet a classifier with perfect robustness and accuracy is still attainable. Concretely, consider the distributions pictured in Figure 14 in Appendix D.10. It is straightforward to show that while there are many perfectly accurate classifiers, any standard loss function will learn an accurate yet non-robust classifier. Only when robust training is employed does the classifier learn a perfectly accurate and perfectly robust decision boundary.

## 5    Related Work

Several models for explaining adversarial examples have been proposed in prior work, utilizing ideas ranging from finite-sample overfitting to high-dimensional statistical phenomena [Gil+18; FFF18; For+19; TG16; Sha+19a; MDM18; Sha+19b; GSS15; BPR18]. The key differentiating aspect of our model is that adversarial perturbations arise as *well-generalizing, yet brittle, features*, rather than statistical anomalies. In particular, adversarial vulnerability does not stem from using a specific model class or a specific training method, since standard training on the "robustified" data distribution of Section 3.1 leads to robust models. At the same time, as shown in Section 3.2, these non-robust features are sufficient to learn a good standard classifier. We discuss the connection between our model and others in detail in Appendix A and additional related work in Appendix B.

## 6    Conclusion

In this work, we cast the phenomenon of adversarial examples as a natural consequence of the presence of *highly predictive but non-robust features* in standard ML datasets. We provide support for this hypothesis by explicitly disentangling robust and non-robust features in standard datasets, as well as showing that non-robust features alone are sufficient for good generalization. Finally, we study these phenomena in more detail in a theoretical setting where we can rigorously study adversarial vulnerability, robust training, and gradient alignment.

Our findings prompt us to view adversarial examples as a fundamentally *human* phenomenon. In particular, we should not be surprised that classifiers exploit highly predictive features that happen to be non-robust under a human-selected notion of similarity, given such features exist in real-world datasets. In the same manner, from the perspective of interpretability, as long as models rely on these non-robust features, we cannot expect to have model explanations that are both human-meaningful and faithful to the models themselves. Overall, attaining models that are robust and interpretable will require explicitly encoding *human priors* into the training process.

## Acknowledgements

We thank Preetum Nakkiran for suggesting the experiment of Appendix D.9 (i.e. replicating Figure 3 with targeted attacks). We also are grateful to the authors of Engstrom et al. [Eng+19a] (Chris Olah, Dan Hendrycks, Justin Gilmer, Reiichiro Nakano, Preetum Nakkiran, Gabriel Goh, Eric Wallace)— for their insights and efforts replicating, extending, and discussing our experimental results.

Work supported in part by the NSF grants CCF-1553428, CCF-1563880, CNS-1413920, CNS-1815221, IIS-1447786, IIS-1607189, the Microsoft Corporation, the Intel Corporation, the MIT-IBM Watson AI Lab research grant, and an Analog Devices Fellowship.

## Footnotes

[2]It is worth emphasizing that while our findings demonstrate that adversarial vulnerability *does* arise from non-robust features, they do not preclude the possibility of adversarial vulnerability also arising from other phenomena [Nak19a]. Nevertheless, the mere existence of useful non-robust features suffices to establish that without explicitly preventing models from utilizing these features, adversarial vulnerability will persist.

[3]The corresponding datasets for CIFAR-10 are publicly available at `http://git.io/adv-datasets`.

[4]In an attempt to explain the gap in accuracy between the model trained on $\widehat{\mathcal{D}}_R$ and the original robust classifier $C$, we test distributional shift, by reporting results on the "robustified" test set in Appendix D.3.

[5]Goh [Goh19a] provides an approach to quantifying this "robust feature leakage" and finds that one can obtain a (small) amount of test accuracy by leveraging robust feature leakage on $\widehat{\mathcal{D}}_{rand}$.

[6]Additional results and analysis are in App. D.5, D.6, and D.7.

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
