[Supplementary Material · supplementary.pdf]

# A   Connections to and Disambiguation from Other Models

Here, we describe other models for adversarial examples and how they relate to the model presented in this paper.

**Concentration of measure in high-dimensions.**   An orthogonal line of work [Gil+18; FFF18; MDM18; Sha+19a], argues that the high dimensionality of the input space can present fundamental barriers on classifier robustness. At a high level, one can show that, for certain data distributions, any decision boundary will be close to a large fraction of inputs and hence no classifier can be robust against small perturbations. While there might exist such fundamental barriers to robustly classifying standard datasets, this model cannot fully explain the situation observed in practice, where one can train (reasonably) robust classifiers on standard datasets [Mad+18; RSL18; WK18; Xia+19; CRK19].

**Insufficient data.**   Schmidt et al. [Sch+18] propose a theoretical model under which a single sample is sufficient to learn a good, yet non-robust classifier, whereas learning a good robust classifier requires $O(\sqrt{d})$ samples. Under this model, adversarial examples arise due to insufficient information about the true data distribution. However, unless the adversary is strong enough (in which case no robust classifier exists), adversarial inputs cannot be utilized as inputs of the opposite class (as done in our experiments in Section 3.2). We note that our model does not explicitly contradict the main thesis of Schmidt et al. [Sch+18]. In fact, this thesis can be viewed as a natural consequence of our conceptual framework. In particular, since training models robustly reduces the effective amount of information in the training data (as non-robust features are discarded), more samples should be required to generalize robustly.

**Boundary Tilting.**   Tanay and Griffin [TG16] introduce the "boundary tilting" model for adversarial examples, and suggest that adversarial examples are a product of over-fitting. In particular, the model conjectures that "adversarial examples are possible because the class boundary extends beyond the submanifold of sample data and can be—under certain circumstances—lying close to it." Consequently, the authors suggest that mitigating adversarial examples may be a matter of regularization and preventing finite-sample overfitting. In contrast, our empirical results in Section 3.2 suggest that adversarial inputs consist of features inherent to the data distribution, since they can encode generalizing information about the target class.

Inspired by this hypothesis and concurrently to our work, Kim, Seo, and Jeon [KSJ19] present a simple classification task comprised of two Gaussian distributions in two dimensions. They experimentally show that the decision boundary tends to better align with the vector between the two means for robust models. This is a special case of our theoretical results in Section 4. (Note that this exact statement is not true beyond two dimensions, as discussed in Section 4.)

**Test Error in Noise.**   Fawzi, Moosavi-Dezfooli, and Frossard [FMF16] and Ford et al. [For+19] argue that the adversarial robustness of a classifier can be directly connected to its robustness under (appropriately scaled) random noise. While this constitutes a natural explanation of adversarial vulnerability given the classifier robustness to noise, these works do not attempt to justify the source of the latter.

At the same time, recent work [Lec+19; CRK19; For+19] utilizes random noise during training or testing to construct adversarially robust classifiers. In the context of our framework, we can expect the added noise to disproportionately affect non-robust features and thus hinder the model's reliance on them.

**Local Linearity.**   Goodfellow, Shlens, and Szegedy [GSS15] suggest that the local linearity of DNNs is largely responsible for the existence of small adversarial perturbations. While this conjecture is supported by the effectiveness of adversarial attacks exploiting local linearity (e.g., FGSM [GSS15]), it is not sufficient to fully characterize the phenomena observed in practice. In particular, there exist adversarial examples that violate the local linearity of the classifier [Mad+18], while classifiers that are less linear do not exhibit greater robustness [ACW18].

**Piecewise-linear decision boundaries.**   Shamir et al. [Sha+19b] prove that the geometric structure of the classifier's decision boundaries can lead to sparse adversarial perturbations. However, this

result does not take into account the distance to the decision boundary along these direction or feasibility constraints on the input domain. As a result, it cannot meaningfully distinguish between classifiers that are brittle to small adversarial perturbations and classifiers that are moderately robust.

**Theoretical constructions which incidentally exploit non-robust features.** Bubeck, Price, and Razenshteyn [BPR18] and Nakkiran [Nak19b] propose theoretical models where the barrier to learning robust classifiers is, respectively, due to computational constraints or model complexity. In order to construct distributions that admit accurate yet non-robust classifiers they (implicitly) utilize the concept of non-robust features. Namely, they add a low-magnitude signal to each input that encodes the true label. This allows a classifier to achieve perfect standard accuracy, but cannot be utilized in an adversarial setting as this signal is susceptible to small adversarial perturbations.

# B    Additional Related Work

We describe previously proposed models for the existence of adversarial examples in the previous section. Here we discuss other work that is methodologically or conceptually similar to ours.

**Distillation.** The experiments performed in Section 3.1 can be seen as a form of *distillation*. There is a line of work, known as model distillation [HVD14; Fur+18; BCN06], where the goal is to train a new model to mimic another already trained model. This is typically achieved by adding some regularization terms to the loss in order to encourage the two models to be similar, often replacing training labels with some other target based on the already trained model. While it might be possible to successfully distill a robust model using these methods, our goal was to achieve it by *only* modifying the training set (leaving the training process unchanged), hence demonstrating that adversarial vulnerability is mainly a property of the dataset. Closer to our work is dataset distillation [Wan+18] which considers the problem of reconstructing a classifier from an alternate dataset much smaller than the original training set. This method aims to produce inputs that directly encode the weights of the already trained model by ensuring that the classifier's gradient with respect to these inputs approximates the desired weights. (As a result, the inputs constructed do not resemble natural inputs.) This approach is orthogonal to our goal since we are not interested in encoding the particular weights into the dataset but rather in imposing a structure to its features.

**Adversarial Transferabiliy.** In our work, we posit that a potentially natural consequence of the existence of non-robust features is *adversarial transferability* [Pap+17; Liu+17; PMG16]. A recent line of work has considered this phenomenon from a theoretical perspective, confined to simple models, or unbounded perturbations [CRP19; Zou+18]. Tramer et al. [Tra+17] study transferability empirically, by finding *adversarial subspaces*, (orthogonal vectors whose linear combinations are adversarial perturbations). The authors find that there is a significant overlap in the adversarial subspaces between different models, and identify this as a source of transferability. In our work, we provide a potential reason for this overlap—these directions correspond to non-robust features utilized by models in a similar manner.

**Universal Adversarial Perturbations** Moosavi-Dezfooli et al. [Moo+17] construct perturbations that can cause misclassification when applied to multiple different inputs. More recently, Jetley, Lord, and Torr [JLT18] discover input patterns that are meaningless to humans and can induce misclassification, while at the same time being essential for standard classification. These findings can be naturally cast into our framework by considering these patterns as non-robust features, providing further evidence about their pervasiveness.

**Manipulating dataset features** Ding et al. [Din+19] perform synthetic transformations on the dataset (e.g., image saturation) and study the performance of models on the transformed dataset under standard and robust training. While this can be seen as a method of restricting the features available to the model during training, it is unclear how well these models would perform on the standard test set. Geirhos et al. [Gei+19] aim to quantify the relative dependence of standard models on shape and texture information of the input. They introduce a version of ImageNet where texture information has been removed and observe an improvement to certain corruptions.

## C  Experimental Setup

### C.1  Datasets

For our experimental analysis, we use the CIFAR-10 [Kri09] and (restricted) ImageNet [Rus+15] datasets. Attaining robust models for the complete ImageNet dataset is known to be a challenging problem, both due to the hardness of the learning problem itself, as well as the computational complexity. We thus restrict our focus to a subset of the dataset which we denote as restricted ImageNet. To this end, we group together semantically similar classes from ImageNet into 9 super-classes shown in Table 2. We train and evaluate only on examples corresponding to these classes.

| Class | Corresponding ImageNet Classes |
|---|---|
| "Dog" | 151 to 268 |
| "Cat" | 281 to 285 |
| "Frog" | 30 to 32 |
| "Turtle" | 33 to 37 |
| "Bird" | 80 to 100 |
| "Primate" | 365 to 382 |
| "Fish" | 389 to 397 |
| "Crab" | 118 to 121 |
| "Insect" | 300 to 319 |

Table 2: Classes used in the Restricted ImageNet model. The class ranges are inclusive.

### C.2  Models

We use the ResNet-50 architecture for our baseline standard and adversarially trained classifiers on CIFAR-10 and restricted ImageNet. For each model, we grid search over three learning rates (0.1, 0.01, 0.05), two batch sizes (128, 256) including/not including a learning rate drop (a single order of magnitude) and data augmentation. We use the standard training parameters for the remaining parameters. The hyperparameters used for each model are given in Table 3.

| Dataset | LR | Batch Size | LR Drop | Data Aug. | Momentum | Weight Decay |
|---|---|---|---|---|---|---|
| $\widehat{\mathcal{D}}_R$ (CIFAR) | 0.1 | 128 | Yes | Yes | 0.9 | $5 \cdot 10^{-4}$ |
| $\widehat{\mathcal{D}}_R$ (Restricted ImageNet) | 0.01 | 128 | No | Yes | 0.9 | $5 \cdot 10^{-4}$ |
| $\widehat{\mathcal{D}}_{NR}$ (CIFAR) | 0.1 | 128 | Yes | Yes | 0.9 | $5 \cdot 10^{-4}$ |
| $\widehat{\mathcal{D}}_{rand}$ (CIFAR) | 0.01 | 128 | Yes | Yes | 0.9 | $5 \cdot 10^{-4}$ |
| $\widehat{\mathcal{D}}_{rand}$ (Restricted ImageNet) | 0.01 | 256 | No | No | 0.9 | $5 \cdot 10^{-4}$ |
| $\widehat{\mathcal{D}}_{det}$ (CIFAR) | 0.1 | 128 | Yes | No | 0.9 | $5 \cdot 10^{-4}$ |
| $\widehat{\mathcal{D}}_{det}$ (Restricted ImageNet) | 0.05 | 256 | No | No | 0.9 | $5 \cdot 10^{-4}$ |

Table 3: Hyperparameters for the models trained in the main paper. All hyperparameters were obtained through a grid search.

## C.3 Adversarial training

To obtain robust classifiers, we employ the adversarial training methodology proposed in [Mad+18]. Specifically, we train against a projected gradient descent (PGD) adversary constrained in $\ell_2$-norm starting from the original image. Following Madry et al. [Mad+18] we normalize the gradient at each step of PGD to ensure that we move a fixed distance in $\ell_2$-norm per step. Unless otherwise specified, we use the values of $\epsilon$ provided in Table 4 to train/evaluate our models. We used 7 steps of PGD with a step size of $\varepsilon/5$.

| Adversary | CIFAR-10 | Restricted Imagenet |
|:---:|:---:|:---:|
| $\ell_2$ | 0.5 | 3 |

Table 4: Value of $\epsilon$ used for $\ell_2$ adversarial training/evaluation of each dataset.

## C.4 Constructing a Robust Dataset

In Section 3.1, we describe a procedure to construct a dataset that contains features relevant only to a given (standard/robust) model. To do so, we optimize the training objective in (5). Unless otherwise specified, we initialize $x_r$ as a different randomly chosen sample from the training set. (For the sake of completeness, we also try initializing with a Gaussian noise instead as shown in Table 7.) We then perform normalized gradient descent ($\ell_2$-norm of gradient is fixed to be constant at each step). At each step we clip the input $x_r$ to in the $[0, 1]$ range so as to ensure that it is a valid image. Details on the optimization procedure are shown in Table 5. We provide the pseudocode for the construction in Figure 5.

GETROBUSTDATASET($D$)

  1. $C_R \leftarrow$ ADVERSARIALTRAINING($D$)
     $g_R \leftarrow$ mapping learned by $C_R$ from the input to the representation layer
  2. $D_R \leftarrow \{\}$
  3. For $(x, y) \in D$
     $x' \sim D$
     $x_R \qquad \leftarrow \qquad \arg\min_{z \in [0,1]^d} \|g_R(z) \quad - \quad g_R(x)\|_2$
     # Solved using $\ell_2$-PGD starting from $x'$
       $D_R \leftarrow D_R \bigcup \{(x_R, y)\}$
  4. Return $D_R$

Figure 5: Algorithm to construct a "robust" dataset, by restricting to features used by a robust model.

|  | CIFAR-10 | Restricted Imagenet |
|:---|:---:|:---:|
| step size | 0.1 | 1 |
| iterations | 1000 | 2000 |

Table 5: Parameters used for optimization procedure to construct dataset in Section 3.1.

## C.5 Non-robust features suffice for standard classification

To construct the dataset as described in Section 3.2, we use the standard projected gradient descent (PGD) procedure described in [Mad+18] to construct an adversarial example for a given input from the dataset (6). Perturbations are constrained in $\ell_2$-norm while each PGD step is normalized to a fixed step size. The details for our PGD setup are described in Table 6. We provide pseudocode in Figure 6.

---

GETNONROBUSTDATASET$(D, \varepsilon)$

    1. $D_{NR} \leftarrow \{\}$

    2. $C \leftarrow$ STANDARDTRAINING$(D)$

    3. For $(x, y) \in D$

        $t \overset{\text{uar}}{\sim} [C]$                # or $t \leftarrow (y + 1) \mod C$

        $x_{NR} \leftarrow \min_{||x'-x|| \leq \varepsilon} L_C(x', t)$   # Solved using $\ell_2$ PGD

        $D_{NR} \leftarrow D_{NR} \bigcup \{(x_{NR}, t)\}$

    4. Return $D_{NR}$

---

Figure 6: Algorithm to construct a dataset where input-label association is based entirely on non-robust features.

| Attack Parameters | CIFAR-10 | Restricted Imagenet |
|:---:|:---:|:---:|
| $\varepsilon$ | 0.5 | 3 |
| step size | 0.1 | 0.1 |
| iterations | 100 | 100 |

Table 6: Projected gradient descent parameters used to construct constrained adversarial examples in Section 3.2.

# D Omitted Experiments and Figures

## D.1 Detailed evaluation of models trained on "robust" dataset

In Section 3.1, we generate a "robust" training set by restricting the dataset to only contain features relevant to a robust model (robust dataset) or a standard model (non-robust dataset). This is performed by choosing either a random input from the training set or random noise[7] and then performing the optimization procedure described in (5). The performance of these classifiers along with various baselines is shown in Table 7. We observe that while the robust dataset constructed from noise resembles the original, the corresponding non-robust does not (Figure 7). This also leads to suboptimal performance of classifiers trained on this dataset (only $46\%$ standard accuracy) potentially due to a distributional shift.

| Model | Accuracy | Robust Accuracy | |
| --- | --- | --- | --- |
| | | $\varepsilon = 0.25$ | $\varepsilon = 0.5$ |
| Standard Training | 95.25 % | 4.49% | 0.0% |
| Robust Training | 90.83% | 82.48% | 70.90% |
| Trained on non-robust dataset (constructed from images) | **87.68%** | 0.82% | 0.0% |
| Trained on non-robust dataset (constructed from noise) | **45.60%** | 1.50% | 0.0% |
| Trained on robust dataset (constructed from images) | 85.40% | **48.20 %** | 21.85% |
| Trained on robust dataset (constructed from noise) | 84.10% | **48.27 %** | 29.40% |

Table 7: Standard and robust classification performance on the CIFAR-10 test set of: an (i) ERM classifier; (ii) ERM classifier trained on a dataset obtained by distilling features relevant to ERM classifier in (i); (iii) adversarially trained classifier ($\varepsilon = 0.5$); (iv) ERM classifier trained on dataset obtained by distilling features used by robust classifier in (iii). Simply restricting the set of available features during ERM to features used by a standard model yields non-trivial robust accuracy.

Figure 7: Robust and non-robust datasets for CIFAR-10 when the process starts from noise (as opposed to random images as in Figure 2a).

## D.2 Adversarial evaluation

To verify the robustness of our classifiers trained on the 'robust' dataset, we evaluate them with strong attacks [Car+19]. In particular, we try up to 2500 steps of projected gradient descent (PGD), increasing steps until the accuracy plateaus, and also try the CW-$\ell_2$ loss function [CW17b] with 1000 steps. For each attack we search over step size. We find that over all attacks and step sizes, the accuracy of the model does not drop by more than 2%, and plateaus at $48.27\%$ for both PGD and CW-$\ell_2$ (the value given in Figure 2). We show a plot of accuracy in terms of the number of PGD steps used in Figure 8.

Figure 8: Robust accuracy as a function of the number of PGD steps used to generate the attack. The accuracy plateaus at $48.27\%$.

## D.3 Performance of "robust" training and test set

In Section 3.1, we observe that an ERM classifier trained on a "robust" training dataset $\widehat{\mathcal{D}}_R$ (obtained by restricting features to those relevant to a robust model) attains non-trivial robustness (cf. Figure 1 and Table 7). In Table 8, we evaluate the adversarial accuracy of the model on the corresponding robust training set (the samples which the classifier was trained on) and test set (unseen samples from $\widehat{\mathcal{D}}_R$, based on the test set). We find that the drop in robustness comes from a combination of generalization gap (the robustness on the $\widehat{\mathcal{D}}_R$ test set is worse than it is on the robust training set) and distributional shift (the model performs better on the robust test set consisting of unseen samples from $\widehat{\mathcal{D}}_R$ than on the standard test set containing unseen samples from $\mathcal{D}$).

| Dataset | Robust Accuracy |
| --- | --- |
| Robust training set | 77.33% |
| Robust test set | 62.49% |
| Standard test set | 48.27% |

Table 8: Performance of model trained on the *robust dataset* on the robust training and test sets as well as the standard CIFAR-10 test set. We observe that the drop in robust accuracy stems from a combination of generalization gap and distributional shift. The adversary is constrained to $\varepsilon = 0.25$ in $\ell_2$-norm.

## D.4 Classification based on non-robust features

Figure 9 shows sample images from $\mathcal{D}$, $\widehat{\mathcal{D}}_{rand}$ and $\widehat{\mathcal{D}}_{det}$ constructed using a standard (non-robust) ERM classifier, and an adversarially trained (robust) classifier.

In Table 9, we repeat the experiments in Table 1 based on datasets constructed using a robust model. Note that using a robust model to generate the $\widehat{\mathcal{D}}_{det}$ and $\widehat{\mathcal{D}}_{rand}$ datasets will not result in non-robust

(a) $\widehat{\mathcal{D}}_{rand}$                               (b) $\widehat{\mathcal{D}}_{det}$

Figure 9: Random samples from datasets where the input-label correlation is entirely based on non-robust features. Samples are generated by performing small adversarial perturbations using either random ($\widehat{\mathcal{D}}_{rand}$) or deterministic ($\widehat{\mathcal{D}}_{det}$) label-target mappings for every sample in the training set. Each image shows: *top*: original; *middle*: adversarial perturbations using a standard ERM-trained classifier; *bottom*: adversarial perturbations using a robust classifier (adversarially trained against $\varepsilon = 0.5$).

features that are strongly predictive of $t$ (since the prediction of the classifier will not change). Thus, training a model on these datasets leads to poor accuracy on the standard test set from $\mathcal{D}$.

Observe from Figure 10 that models trained on datasets derived from the robust model show a decline in test accuracy as training progresses. In Table 9, the accuracy numbers reported correspond to the *last* iteration, and not the *best* performance. This is because we have no way to cross-validate in a meaningful way as the validation set itself comes from $\widehat{\mathcal{D}}_{rand}$ or $\widehat{\mathcal{D}}_{det}$, and not from the true data distribution $D$. Thus, validation accuracy will not be predictive of the true test accuracy, and thus will not help determine when to early stop.

| Model used to construct dataset | Dataset used in training | | |
|---|---|---|---|
| | $\mathcal{D}$ | $\widehat{\mathcal{D}}_{rand}$ | $\widehat{\mathcal{D}}_{det}$ |
| Robust | 95.3% | 25.2 % | 5.8% |
| Standard | 95.3% | 63.3 % | 43.7% |

Table 9: Repeating the experiments of Table 1 using a robust model to construct the datasets $\mathcal{D}$, $\widehat{\mathcal{D}}_{rand}$ and $\widehat{\mathcal{D}}_{det}$. Results in Table 1 are reiterated for comparison.

## D.5 Accuracy curves

(a) Trained using $\widehat{\mathcal{D}}_{rand}$ training set

(b) Trained using $\widehat{\mathcal{D}}_{det}$ training set

Figure 10: Test accuracy on $\mathcal{D}$ of standard classifiers trained on datasets where input-label correlation is based solely on non-robust features as in Section 3.2. The datasets are constructed using either a non-robust/standard model (*left column*) or a robust model (*right column*). The labels used are either random ($\widehat{\mathcal{D}}_{rand}$; *top row*) or correspond to a deterministic permutation ($\widehat{\mathcal{D}}_{det}$; *bottom row*).

## D.6 Performance of ERM classifiers on relabeled test set

In Table 10), we evaluate the performance of classifiers trained on $\widehat{\mathcal{D}}_{det}$ on both the original test set drawn from $\mathcal{D}$, and the test set relabelled using $t(y) = (y+1) \mod C$. Observe that the classifier trained on $\widehat{\mathcal{D}}_{det}$ constructed using a robust model actually ends up learning permuted labels based on robust features (indicated by high test accuracy on the relabelled test set).

| Model used to construct training dataset for $\widehat{\mathcal{D}}_{det}$ | Dataset used in testing | |
|---|---|---|
| | $\mathcal{D}$ | relabelled-$\mathcal{D}$ |
| Standard | 43.7% | 16.2% |
| Robust | 5.8% | 65.5% |

Table 10: Performance of classifiers trained using $\widehat{\mathcal{D}}_{det}$ training set constructed using either standard or robust models. The classifiers are evaluated both on the standard test set from $\mathcal{D}$ and the test set relabeled using $t(y) = (y+1) \mod C$. We observe that using a robust model for the construction results in a model that largely predicts the permutation of labels, indicating that the dataset does not have strongly predictive non-robust features.

## D.7 Generalization to CIFAR-10.1

Recht et al. [Rec+19] have constructed an unseen but distribution-shifted test set for CIFAR-10. They show that for many previously proposed models, accuracy on the CIFAR-10.1 test set can be predicted as a linear function of performance on the CIFAR-10 test set.

As a sanity check (and a safeguard against any potential adaptive overfitting to the test set via hyperparameters, historical test set reuse, etc.) we note that the classifiers trained on $\widehat{\mathcal{D}}_{det}$ and $\widehat{\mathcal{D}}_{rand}$ achieve $44\%$ and $55\%$ generalization on the CIFAR-10.1 test set, respectively. This demonstrates non-trivial generalization, and actually perform better than the linear fit would predict (given their accuracies on the CIFAR-10 test set).

## D.8 Omitted Results for Restricted ImageNet

Figure 11: Repeating the experiments shown in Figure 2 for the Restricted ImageNet dataset. Sample images from the resulting dataset.

Figure 12: Repeating the experiments shown in Figure 2 for the Restricted ImageNet dataset. Standard and robust accuracy of models trained on these datasets.

## D.9 Targeted Transferability

Figure 13: Transfer rate of *targeted* adversarial examples (measured in terms of attack success rate, not just misclassification) from a ResNet-50 to different architectures alongside test set performance of these architecture when trained on the dataset generated in Section 3.2. Architectures more susceptible to transfer attacks also performed better on the standard test set supporting our hypothesis that adversarial transferability arises from utilizing similar *non-robust features*.

## D.10 Robustness vs. Accuracy

Figure 14: An example where adversarial vulnerability can arise from ERM training on any standard loss function due to non-robust features (the green line shows the ERM-learned decision boundary). There exists, however, a classifier that is both perfectly robust *and* accurate, resulting from robust training, which forces the classifier to ignore the $x_2$ feature despite its predictiveness.

# E   Gaussian MLE under Adversarial Perturbation

In this section, we develop a framework for studying non-robust features by studying the problem of *maximum likelihood classification* between two Gaussian distributions. We first recall the setup of the problem, then present the main theorems from Section 4. First we build the techniques necessary for their proofs.

## E.1   Setup

We consider the setup where a learner receives labeled samples from two distributions, $\mathcal{N}(\boldsymbol{\mu}_*, \boldsymbol{\Sigma}_*)$, and $\mathcal{N}(-\boldsymbol{\mu}_*, \boldsymbol{\Sigma}_*)$. The learner's goal is to be able to classify new samples as being drawn from $\mathcal{D}_1$ or $\mathcal{D}_2$ according to a maximum likelihood (MLE) rule.

A simple coupling argument demonstrates that this problem can actually be reduced to learning the parameters $\widehat{\boldsymbol{\mu}}$, $\widehat{\boldsymbol{\Sigma}}$ of a single Gaussian $\mathcal{N}(-\boldsymbol{\mu}_*, \boldsymbol{\Sigma}_*)$, and then employing a linear classifier with weight $\widehat{\boldsymbol{\Sigma}}^{-1}\widehat{\boldsymbol{\mu}}$. In the standard setting, maximum likelihoods estimation learns the true parameters, $\boldsymbol{\mu}_*$ and $\boldsymbol{\Sigma}_*$, and thus the learned classification rule is $C(x) = \mathbb{1}\{x^\top \boldsymbol{\Sigma}^{-1}\boldsymbol{\mu} > 0\}$.

In this work, we consider the problem of *adversarially robust* maximum likelihood estimation. In particular, rather than simply being asked to classify samples, the learner will be asked to classify *adversarially perturbed* samples $x + \delta$, where $\delta \in \Delta$ is chosen to maximize the loss of the learner. Our goal is to derive the parameters $\boldsymbol{\mu}, \boldsymbol{\Sigma}$ corresponding to an adversarially robust maximum likelihood estimate of the parameters of $\mathcal{N}(\boldsymbol{\mu}_*, \boldsymbol{\Sigma}_*)$. Note that since we have access to $\boldsymbol{\Sigma}_*$ (indeed, the learner can just run non-robust MLE to get access), we work in the space where $\boldsymbol{\Sigma}^*$ is a diagonal matrix, and we restrict the learned covariance $\boldsymbol{\Sigma}$ to the set of diagonal matrices.

**Notation.**   We denote the parameters of the sampled Gaussian by $\boldsymbol{\mu}_* \in \mathbb{R}^d$, and $\boldsymbol{\Sigma}_* \in \{\mathrm{diag}(\boldsymbol{u})|\boldsymbol{u} \in \mathbb{R}^d\}$. We use $\sigma_{min}(X)$ to represent the smallest eigenvalue of a square matrix $X$, and $\ell(\cdot; x)$ to represent the Gaussian negative log-likelihood for a single sample $x$. For convenience, we often use $\boldsymbol{v} = x - \boldsymbol{\mu}$, and $R = \|\boldsymbol{\mu}_*\|$. We also define the $\diagdown$ operator to represent the vectorization of the diagonal of a matrix. In particular, for a matrix $X \in \mathbb{R}^{d \times d}$, we have that $X_{\diagdown} = v \in \mathbb{R}^d$ if $v_i = X_{ii}$.

## E.2   Outline and Key Results

We focus on the case where $\Delta = \mathcal{B}_2(\epsilon)$ for some $\epsilon > 0$, i.e. the $\ell_2$ ball, corresponding to the following minimax problem:

$$\min_{\boldsymbol{\mu}, \boldsymbol{\Sigma}} \mathbb{E}_{x \sim \mathcal{N}(\boldsymbol{\mu}^*, \boldsymbol{\Sigma}^*)} \left[ \max_{\delta: \|\delta\| = \varepsilon} \ell(\boldsymbol{\mu}, \boldsymbol{\Sigma}; x + \delta) \right] \tag{12}$$

We first derive the optimal adversarial perturbation for this setting (Section E.3.1), and prove Theorem 1 (Section E.3.2). We then propose an alternate problem, in which the adversary picks a linear operator to be applied to a fixed vector, rather than picking a specific perturbation vector (Section E.3.3). We argue via Gaussian concentration that the alternate problem is indeed reflective of the original model (and in particular, the two become equivalent as $d \to \infty$). In particular, we propose studying the following in place of (12):

$$\min_{\boldsymbol{\mu}, \boldsymbol{\Sigma}} \max_{M \in \mathcal{M}} \mathbb{E}_{x \sim \mathcal{N}(\boldsymbol{\mu}^*, \boldsymbol{\Sigma}^*)} \left[ \ell(\boldsymbol{\mu}, \boldsymbol{\Sigma}; x + M(x - \boldsymbol{\mu})) \right] \tag{13}$$

$$\text{where } \mathcal{M} = \left\{ M \in \mathbb{R}^{d \times d} : \; M_{ij} = 0 \; \forall \; i \neq j, \; \mathbb{E}_{x \sim \mathcal{N}(\boldsymbol{\mu}^*, \boldsymbol{\Sigma}^*)} \left[ \|M\boldsymbol{v}\|_2^2 \right] = \epsilon^2 \right\}.$$

Our goal is to characterize the behavior of the robustly learned covariance $\boldsymbol{\Sigma}$ in terms of the true covariance matrix $\boldsymbol{\Sigma}_*$ and the perturbation budget $\varepsilon$. The proof is through Danskin's Theorem, which allows us to use any maximizer of the inner problem $M^*$ in computing the subgradient of the inner minimization. After showing the applicability of Danskin's Theorem (Section E.3.4) and then applying it (Section E.3.5) to prove our main results (Section E.3.7). Our three main results, which we prove in the following section, are presented below.

First, we consider a simplified version of (12), in which the adversary solves a maximization with a fixed Lagrangian penalty, rather than a hard $\ell_2$ constraint. In this setting, we show that the loss contributed by the adversary corresponds to a misalignment between the data metric (the Mahalanobis distance, induced by $\boldsymbol{\Sigma}^{-1}$), and the $\ell_2$ metric:

**Theorem 1** (Adversarial vulnerability from misalignment). *Consider an adversary whose perturbation is determined by the "Lagrangian penalty" form of (11), i.e.*

$$\max_{\delta} \ell(x + \delta; y \cdot \boldsymbol{\mu}, \boldsymbol{\Sigma}) - C \cdot \|\delta\|_2,$$

*where $C \geq \frac{1}{\sigma_{min}(\boldsymbol{\Sigma}_*)}$ is a constant trading off NLL minimization and the adversarial constraint (the bound on $C$ ensures the problem is concave). Then, the adversarial loss $\mathcal{L}_{adv}$ incurred by $(\boldsymbol{\mu}, \boldsymbol{\Sigma})$ is*

$$\mathcal{L}_{adv}(\Theta) - \mathcal{L}(\Theta) = tr\left[\left(I + (C \cdot \boldsymbol{\Sigma}_* - I)^{-1}\right)^2\right] - d,$$

*and, for a fixed $tr(\boldsymbol{\Sigma}_*) = k$ the above is minimized by $\boldsymbol{\Sigma}_* = \frac{k}{d}\boldsymbol{I}$.*

We then return to studying (13), where we provide upper and lower bounds on the learned robust covariance matrix $\boldsymbol{\Sigma}$:

**Theorem 2** (Robustly Learned Parameters). *Just as in the non-robust case, $\boldsymbol{\mu}_r = \boldsymbol{\mu}^*$, i.e. the true mean is learned. For the robust covariance $\boldsymbol{\Sigma}_r$, there exists an $\varepsilon_0 > 0$, such that for any $\varepsilon \in [0, \varepsilon_0)$,*

$$\boldsymbol{\Sigma}_r = \frac{1}{2}\boldsymbol{\Sigma}_* + \frac{1}{\lambda} \cdot \boldsymbol{I} + \sqrt{\frac{1}{\lambda} \cdot \boldsymbol{\Sigma}_* + \frac{1}{4}\boldsymbol{\Sigma}_*^2}, \qquad \text{where} \qquad \Omega\left(\frac{1 + \varepsilon^{1/2}}{\varepsilon^{1/2} + \varepsilon^{3/2}}\right) \leq \lambda \leq O\left(\frac{1 + \varepsilon^{1/2}}{\varepsilon^{1/2}}\right).$$

Finally, we show that in the worst case over mean vectors $\boldsymbol{\mu}_*$, the gradient of the adversarial robust classifier aligns more with the inter-class vector:

**Theorem 3** (Gradient alignment). *Let $f(x)$ and $f_r(x)$ be monotonic classifiers based on the linear separator induced by standard and $\ell_2$-robust maximum likelihood classification, respectively. The maximum angle formed between the gradient of the classifier (wrt input) and the vector connecting the classes can be smaller for the robust model:*

$$\min_{\boldsymbol{\mu}} \frac{\langle \boldsymbol{\mu}, \nabla_x f_r(x)\rangle}{\|\boldsymbol{\mu}\| \cdot \|\nabla_x f_r(x)\|} > \min_{\boldsymbol{\mu}} \frac{\langle \boldsymbol{\mu}, \nabla_x f(x)\rangle}{\|\boldsymbol{\mu}\| \cdot \|\nabla_x f(x)\|}.$$

### E.3 Proofs

In the first section, we have shown that the classification between two Gaussian distributions with identical covariance matrices centered at $\boldsymbol{\mu}^*$ and $-\boldsymbol{\mu}^*$ can in fact be reduced to learning the parameters of a single one of these distributions.

Thus, in the standard setting, our goal is to solve the following problem:

$$\min_{\boldsymbol{\mu}, \boldsymbol{\Sigma}} \mathbb{E}_{x \sim \mathcal{N}(\boldsymbol{\mu}^*, \boldsymbol{\Sigma}^*)}\left[\ell(\boldsymbol{\mu}, \boldsymbol{\Sigma}; x)\right] := \min_{\boldsymbol{\mu}, \boldsymbol{\Sigma}} \mathbb{E}_{x \sim \mathcal{N}(\boldsymbol{\mu}^*, \boldsymbol{\Sigma}^*)}\left[-\log\left(\mathcal{N}(\boldsymbol{\mu}, \boldsymbol{\Sigma}; x)\right)\right].$$

Note that in this setting, one can simply find differentiate $\ell$ with respect to both $\boldsymbol{\mu}$ and $\boldsymbol{\Sigma}$, and obtain closed forms for both (indeed, these closed forms are, unsurprisingly, $\boldsymbol{\mu}^*$ and $\boldsymbol{\Sigma}^*$). Here, we consider the existence of a *malicious adversary* who is allowed to perturb each sample point $x$ by some $\delta$. The goal of the adversary is to *maximize* the same loss that the learner is minimizing.

#### E.3.1 Motivating example: $\ell_2$-constrained adversary

We first consider, as a motivating example, an $\ell_2$-constrained adversary. That is, the adversary is allowed to perturb each sampled point by $\delta : \|\delta\|_2 = \varepsilon$. In this case, the minimax problem being solved is the following:

$$\min_{\boldsymbol{\mu}, \boldsymbol{\Sigma}} \mathbb{E}_{x \sim \mathcal{N}(\boldsymbol{\mu}^*, \boldsymbol{\Sigma}^*)}\left[\max_{\|\delta\| = \varepsilon} \ell(\boldsymbol{\mu}, \boldsymbol{\Sigma}; x + \delta)\right]. \tag{14}$$

The following Lemma captures the optimal behaviour of the adversary:

**Lemma 1.** *In the minimax problem captured in (14) (and earlier in (12)), the optimal adversarial perturbation $\delta^*$ is given by*

$$\delta^* = \left(\lambda \boldsymbol{I} - \boldsymbol{\Sigma}^{-1}\right)^{-1} \boldsymbol{\Sigma}^{-1}\boldsymbol{v} = \left(\lambda \boldsymbol{\Sigma} - \boldsymbol{I}\right)^{-1} \boldsymbol{v}, \tag{15}$$

*where $\boldsymbol{v} = x - \boldsymbol{\mu}$, and $\lambda$ is set such that $\|\delta^*\|_2 = \varepsilon$.*

*Proof.* In this context, we can solve the inner maximization problem with Lagrange multipliers. In the following we write $\Delta = \mathcal{B}_2(\varepsilon)$ for brevity, and discard terms not containing $\delta$ as well as constant factors freely:

$$
\begin{aligned}
\arg\max_{\delta \in \Delta} \ell(\boldsymbol{\mu}, \boldsymbol{\Sigma}; x + \delta) - &= \arg\max_{\delta \in \Delta} (x + \delta - \boldsymbol{\mu})^\top \boldsymbol{\Sigma}^{-1} (x + \delta - \boldsymbol{\mu}) \\
&= \arg\max_{\delta \in \Delta} (x - \boldsymbol{\mu})^\top \boldsymbol{\Sigma}^{-1} (x - \boldsymbol{\mu}) + 2\delta^\top \boldsymbol{\Sigma}^{-1}(x - \boldsymbol{\mu}) + \delta^\top \boldsymbol{\Sigma}^{-1}\delta \\
&= \arg\max_{\delta \in \Delta} \delta^\top \boldsymbol{\Sigma}^{-1}(x - \boldsymbol{\mu}) + \frac{1}{2}\delta^\top \boldsymbol{\Sigma}^{-1}\delta. \quad (16)
\end{aligned}
$$

Now we can solve (16) using the aforementioned Lagrange multipliers. In particular, note that the maximum of (16) is attained at the boundary of the $\ell_2$ ball $\Delta$. Thus, we can solve the following system of two equations to find $\delta$, rewriting the norm constraint as $\frac{1}{2}\|\delta\|_2^2 = \frac{1}{2}\varepsilon^2$:

$$
\begin{cases}
\nabla_\delta \left( \delta^\top \boldsymbol{\Sigma}^{-1}(x - \boldsymbol{\mu}) + \frac{1}{2}\delta^\top \boldsymbol{\Sigma}^{-1}\delta \right) = \lambda \nabla_\delta \left( \|\delta\|_2^2 - \varepsilon^2 \right) \implies \boldsymbol{\Sigma}^{-1}(x - \boldsymbol{\mu}) + \boldsymbol{\Sigma}^{-1}\delta = \lambda\delta \\
\|\delta\|_2^2 = \varepsilon^2.
\end{cases}
$$
$$(17)$$

For clarity, we write $\boldsymbol{v} = x - \boldsymbol{\mu}$: then, combining the above, we have that

$$
\delta^* = \left(\lambda \boldsymbol{I} - \boldsymbol{\Sigma}^{-1}\right)^{-1} \boldsymbol{\Sigma}^{-1}\boldsymbol{v} = \left(\lambda\boldsymbol{\Sigma} - \boldsymbol{I}\right)^{-1}\boldsymbol{v}, \quad (18)
$$

our final result for the maximizer of the inner problem, where $\lambda$ is set according to the norm constraint. $\qquad\square$

### E.3.2 Variant with Fixed Lagrangian (Theorem 1)

To simplify the analysis of Theorem 1, we consider a version of (14) with a fixed Lagrangian penalty, rather than a norm constraint:

$$
\max \ell(x + \delta; y \cdot \boldsymbol{\mu}, \boldsymbol{\Sigma}) - C \cdot \|\delta\|_2.
$$

Note then, that by Lemma 1, the optimal perturbation $\delta^*$ is given by

$$
\delta^* = (C\boldsymbol{\Sigma} - \boldsymbol{I})^{-1}.
$$

We now proceed to the proof of Theorem 1.

**Theorem 1** (Adversarial vulnerability from misalignment)**.** *Consider an adversary whose perturbation is determined by the "Lagrangian penalty" form of* (11)*, i.e.*

$$
\max_\delta \ell(x + \delta; y \cdot \boldsymbol{\mu}, \boldsymbol{\Sigma}) - C \cdot \|\delta\|_2,
$$

*where $C \geq \frac{1}{\sigma_{min}(\boldsymbol{\Sigma}_*)}$ is a constant trading off NLL minimization and the adversarial constraint (the bound on $C$ ensures the problem is concave). Then, the adversarial loss $\mathcal{L}_{adv}$ incurred by $(\boldsymbol{\mu}, \boldsymbol{\Sigma})$ is*

$$
\mathcal{L}_{adv}(\Theta) - \mathcal{L}(\Theta) = tr\left[ \left( I + (C \cdot \boldsymbol{\Sigma}_* - I)^{-1} \right)^2 \right] - d,
$$

*and, for a fixed $tr(\boldsymbol{\Sigma}_*) = k$ the above is minimized by $\boldsymbol{\Sigma}_* = \frac{k}{d}\boldsymbol{I}$.*

*Proof.* We begin by expanding the Gaussian negative log-likelihood for the relaxed problem:

$$
\begin{aligned}
\mathcal{L}_{adv}(\Theta) - \mathcal{L}(\Theta) &= \mathbb{E}_{x \sim \mathcal{N}(\boldsymbol{\mu}^*, \boldsymbol{\Sigma}^*)} \left[ 2 \cdot \boldsymbol{v}^\top (C \cdot \boldsymbol{\Sigma} - \boldsymbol{I})^{-\top} \boldsymbol{\Sigma}^{-1}\boldsymbol{v} + \boldsymbol{v}^\top (C \cdot \boldsymbol{\Sigma} - \boldsymbol{I})^{-\top} \boldsymbol{\Sigma}^{-1} (C \cdot \boldsymbol{\Sigma} - \boldsymbol{I})^{-1} \boldsymbol{v} \right] \\
&= \mathbb{E}_{x \sim \mathcal{N}(\boldsymbol{\mu}^*, \boldsymbol{\Sigma}^*)} \left[ 2 \cdot \boldsymbol{v}^\top (C \cdot \boldsymbol{\Sigma}\boldsymbol{\Sigma} - \boldsymbol{\Sigma})^{-1} \boldsymbol{v} + \boldsymbol{v}^\top (C \cdot \boldsymbol{\Sigma} - \boldsymbol{I})^{-\top} \boldsymbol{\Sigma}^{-1} (C \cdot \boldsymbol{\Sigma} - \boldsymbol{I})^{-1} \boldsymbol{v} \right]
\end{aligned}
$$

Recall that we are considering the vulnerability at the MLE parameters $\boldsymbol{\mu}^*$ and $\boldsymbol{\Sigma}^*$:

$$\mathcal{L}_{adv}(\Theta) - \mathcal{L}(\Theta) = \mathbb{E}_{\boldsymbol{v} \sim \mathcal{N}(0,I)} \left[ 2 \cdot \boldsymbol{v}^\top \boldsymbol{\Sigma}_*^{1/2} \left( C \cdot \boldsymbol{\Sigma}_*^2 - \boldsymbol{\Sigma}_* \right)^{-1} \boldsymbol{\Sigma}_*^{1/2} \boldsymbol{v} \right.$$
$$\left. + \boldsymbol{v}^\top \boldsymbol{\Sigma}_*^{1/2} \left( C \cdot \boldsymbol{\Sigma}_* - \boldsymbol{I} \right)^{-\top} \boldsymbol{\Sigma}_*^{-1} \left( C \cdot \boldsymbol{\Sigma}_* - \boldsymbol{I} \right)^{-1} \boldsymbol{\Sigma}_*^{1/2} \boldsymbol{v} \right]$$
$$= \mathbb{E}_{\boldsymbol{v} \sim \mathcal{N}(0,I)} \left[ 2 \cdot \boldsymbol{v}^\top \left( C \cdot \boldsymbol{\Sigma}_* - \boldsymbol{I} \right)^{-1} \boldsymbol{v} + \boldsymbol{v}^\top \boldsymbol{\Sigma}_*^{1/2} \left( C^2 \boldsymbol{\Sigma}_*^3 - 2C \cdot \boldsymbol{\Sigma}_*^2 + \boldsymbol{\Sigma}_* \right)^{-1} \boldsymbol{\Sigma}_*^{1/2} \boldsymbol{v} \right]$$
$$= \mathbb{E}_{\boldsymbol{v} \sim \mathcal{N}(0,I)} \left[ 2 \cdot \boldsymbol{v}^\top \left( C \cdot \boldsymbol{\Sigma}_* - \boldsymbol{I} \right)^{-1} \boldsymbol{v} + \boldsymbol{v}^\top \left( C \cdot \boldsymbol{\Sigma}_* - \boldsymbol{I} \right)^{-2} \boldsymbol{v} \right]$$
$$= \mathbb{E}_{\boldsymbol{v} \sim \mathcal{N}(0,I)} \left[ -\|v\|_2^2 + \boldsymbol{v}^\top \boldsymbol{I} \boldsymbol{v} + 2 \cdot \boldsymbol{v}^\top \left( C \cdot \boldsymbol{\Sigma}_* - \boldsymbol{I} \right)^{-1} \boldsymbol{v} + \boldsymbol{v}^\top \left( C \cdot \boldsymbol{\Sigma}_* - \boldsymbol{I} \right)^{-2} \boldsymbol{v} \right]$$
$$= \mathbb{E}_{\boldsymbol{v} \sim \mathcal{N}(0,I)} \left[ -\|v\|_2^2 + \boldsymbol{v}^\top \left( \boldsymbol{I} + (C \cdot \boldsymbol{\Sigma}_* - \boldsymbol{I})^{-1} \right)^2 \boldsymbol{v} \right]$$
$$= \mathrm{tr} \left[ \left( \boldsymbol{I} + (C \cdot \boldsymbol{\Sigma}_* - \boldsymbol{I})^{-1} \right)^2 \right] - d$$

This shows the first part of the theorem. It remains to show that for a fixed $k = \mathrm{tr}(\boldsymbol{\Sigma}_*)$, the adversarial risk is minimized by $\boldsymbol{\Sigma}_* = \frac{k}{d} \boldsymbol{I}$:

$$\min_{\boldsymbol{\Sigma}_*} \mathcal{L}_{adv}(\Theta) - \mathcal{L}(\Theta) = \min_{\boldsymbol{\Sigma}_*} \mathrm{tr} \left[ \left( \boldsymbol{I} + (C \cdot \boldsymbol{\Sigma}_* - \boldsymbol{I})^{-1} \right)^2 \right]$$
$$= \min_{\{\sigma_i\}} \sum_{i=1}^{d} \left( 1 + \frac{1}{C \cdot \sigma_i - 1} \right)^2 ,$$

where $\{\sigma_i\}$ are the eigenvalues of $\boldsymbol{\Sigma}_*$. Now, we have that $\sum \sigma_i = k$ by assumption, so by optimality conditions, we have that $\boldsymbol{\Sigma}_*$ minimizes the above if $\nabla_{\{\sigma_i\}} \propto \vec{1}$, i.e. if $\nabla_{\sigma_i} = \nabla_{\sigma_j}$ for all $i, j$. Now,

$$\nabla_{\sigma_i} = -2 \cdot \left( 1 + \frac{1}{C \cdot \sigma_i - 1} \right) \cdot \frac{C}{(C \cdot \sigma_i - 1)^2}$$
$$= -2 \cdot \frac{C^2 \cdot \sigma_i}{(C \cdot \sigma_i - 1)^3}.$$

Then, by solving analytically, we find that

$$-2 \cdot \frac{C^2 \cdot \sigma_i}{(C \cdot \sigma_i - 1)^3} = -2 \cdot \frac{C^2 \cdot \sigma_j}{(C \cdot \sigma_j - 1)^3}$$

admits only one real solution, $\sigma_i = \sigma_j$. Thus, $\boldsymbol{\Sigma}_* \propto \boldsymbol{I}$. Scaling to satisfy the trace constraint yields $\boldsymbol{\Sigma}_* = \frac{k}{d} \boldsymbol{I}$, which concludes the proof. $\qquad\square$

### E.3.3 Real objective

Our motivating example (Section E.3.1) demonstrates that the optimal perturbation for the adversary in the $\ell_2$-constrained case is actually a linear function of $\boldsymbol{v}$, and in particular, that the optimal perturbation can be expressed as $D\boldsymbol{v}$ for a diagonal matrix $D$. Note, however, that the problem posed in (14) is not actually a minimax problem, due to the presence of the expectation between the outer minimization and the inner maximization. Motivated by this and (18), we define the following robust problem:

$$\min_{\boldsymbol{\mu}, \boldsymbol{\Sigma}} \max_{M \in \mathcal{M}} \mathbb{E}_{x \sim \mathcal{N}(\boldsymbol{\mu}^*, \boldsymbol{\Sigma}^*)} \left[ \ell(\boldsymbol{\mu}, \boldsymbol{\Sigma}; x + M\boldsymbol{v}) \right], \tag{19}$$

$$\text{where } \mathcal{M} = \left\{ M \in \mathbb{R}^{d \times d} : M_{ij} = 0 \ \forall \ i \neq j, \ \mathbb{E}_{x \sim \mathcal{N}(\boldsymbol{\mu}^*, \boldsymbol{\Sigma}^*)} \left[ \|M\boldsymbol{v}\|_2^2 \right] = \epsilon^2 \right\}.$$

First, note that this objective is slightly different from that of (14). In the motivating example, $\delta$ is constrained to *always* have $\varepsilon$-norm, and thus is normalizer on a per-sample basis inside of the expectation. In contrast, here the classifier is concerned with being robust to perturbations that are linear in $\boldsymbol{v}$, and of $\varepsilon^2$ squared norm *in expectation*.

Note, however, that via the result of Laurent and Massart [LM00] showing strong concentration for the norms of Gaussian random variables, in high dimensions this bound on expectation has a corresponding high-probability bound on the norm. In particular, this implies that as $d \rightarrow \infty$, $\|Mv\|_2 = \varepsilon$ almost surely, and thus the problem becomes identical to that of (14). We now derive the optimal $M$ for a given $(\mu, \Sigma)$:

**Lemma 2.** *Consider the minimax problem described by* (19), *i.e.*

$$\min_{\mu, \Sigma} \max_{M \in \mathcal{M}} \mathbb{E}_{x \sim \mathcal{N}(\mu^*, \Sigma^*)} \left[ \ell(\mu, \Sigma; x + Mv) \right].$$

*Then, the optimal action $M^*$ of the inner maximization problem is given by*

$$M = (\lambda \Sigma - I)^{-1}, \tag{20}$$

*where again $\lambda$ is set so that $M \in \mathcal{M}$.*

*Proof.* We accomplish this in a similar fashion to what was done for $\delta^*$, using Lagrange multipliers:

$$\nabla_M \mathbb{E}_{x \sim \mathcal{N}(\mu^*, \Sigma^*)} \left[ v^\top M \Sigma^{-1} v + \frac{1}{2} v^\top M \Sigma^{-1} M v \right] = \lambda \nabla_M \mathbb{E}_{x \sim \mathcal{N}(\mu^*, \Sigma^*)} \left[ \|Mv\|_2^2 - \varepsilon^2 \right]$$

$$\mathbb{E}_{x \sim \mathcal{N}(\mu^*, \Sigma^*)} \left[ \Sigma^{-1} vv^\top + \Sigma^{-1} M vv^\top \right] = \mathbb{E}_{x \sim \mathcal{N}(\mu^*, \Sigma^*)} \left[ \lambda M vv^\top \right]$$

$$\Sigma^{-1} \Sigma^* + \Sigma^{-1} M \Sigma^* = \lambda M \Sigma^*$$

$$M = (\lambda \Sigma - I)^{-1},$$

where $\lambda$ is a constant depending on $\Sigma$ and $\mu$ enforcing the expected squared-norm constraint. $\square$

Indeed, note that the optimal $M$ for the adversary takes a near-identical form to the optimal $\delta$ (18), with the exception that $\lambda$ is not sample-dependent but rather varies only with the parameters.

### E.3.4 Danskin's Theorem

The main tool in proving our key results is Danskin's Theorem [Dan67], a powerful theorem from minimax optimization which contains the following key result:

**Theorem 4** (Danskin's Theorem). *Suppose $\phi(x, z) : \mathbb{R} \times Z \rightarrow \mathbb{R}$ is a continuous function of two arguments, where $Z \subset \mathbb{R}^m$ is compact. Define $f(x) = \max_{z \in Z} \phi(x, z)$. Then, if for every $z \in Z$, $\phi(x, z)$ is convex and differentiable in $x$, and $\frac{\partial \phi}{\partial x}$ is continuous:*

*The subdifferential of $f(x)$ is given by*

$$\partial f(x) = \text{conv} \left\{ \frac{\partial \phi(x, z)}{\partial x} : z \in Z_0(x) \right\},$$

*where $\text{conv}(\cdot)$ represents the convex hull operation, and $Z_0$ is the set of maximizers defined as*

$$Z_0(x) = \left\{ \overline{z} : \phi(x, \overline{z}) = \max_{z \in Z} \phi(x, z) \right\}.$$

In short, given a minimax problem of the form $\min_x \max_{y \in C} f(x, y)$ where $C$ is a compact set, if $f(\cdot, y)$ is convex for all values of $y$, then rather than compute the gradient of $g(x) := \max_{y \in C} f(x, y)$, we can simply find a maximizer $y^*$ for the current parameter $x$; Theorem 4 ensures that $\nabla_x f(x, y^*) \in \partial_x g(x)$. Note that $\mathcal{M}$ is trivially compact (by the Heine-Borel theorem), and differentiability/continuity follow rather straightforwardly from our reparameterization (c.f. (21)), and so it remains to show that the outer minimization is convex for any fixed $M$.

**Convexity of the outer minimization.** Note that even in the standard case (i.e. non-adversarial), the Gaussian negative log-likelihood is not convex with respect to $(\mu, \Sigma)$. Thus, rather than proving convexity of this function directly, we employ the parameterization used by [Das+19]: in particular, we write the problem in terms of $T = \Sigma^{-1}$ and $m = \Sigma^{-1} \mu$. Under this parameterization, we show that the robust problem is convex for any fixed $M$.

**Lemma 3.** *Under the aforementioned parameterization of $\boldsymbol{T} = \boldsymbol{\Sigma}^{-1}$ and $\boldsymbol{m} = \boldsymbol{\Sigma}^{-1}\boldsymbol{\mu}$, the following "Gaussian robust negative log-likelihood" is convex:*

$$\mathbb{E}_{x \sim \mathcal{N}(\boldsymbol{\mu}^*, \boldsymbol{\Sigma}^*)} \left[ \ell(\boldsymbol{m}, \boldsymbol{T}; x + M\boldsymbol{v}) \right].$$

*Proof.* To prove this, we show that the likelihood is convex even with respect to a single sample $x$; the result follows, since a convex combination of convex functions remains convex. We begin by looking at the likelihood of a single sample $x \sim \mathcal{N}(\boldsymbol{\mu}_*, \boldsymbol{\Sigma}_*)$:

$$\mathcal{L}(\boldsymbol{\mu}, \boldsymbol{\Sigma}; x + M(x - \boldsymbol{\mu})) = \frac{1}{\sqrt{(2\pi)^k |\boldsymbol{\Sigma}|}} \exp\left( -\frac{1}{2}(x - \boldsymbol{\mu})^\top (I + M)^2 \boldsymbol{\Sigma}^{-1}(x - \boldsymbol{\mu}) \right)$$

$$= \frac{\frac{1}{\sqrt{(2\pi)^k |\boldsymbol{\Sigma}|}} \exp\left( -\frac{1}{2}(x - \boldsymbol{\mu})^\top (I + M)^2 \boldsymbol{\Sigma}^{-1}(x - \boldsymbol{\mu}) \right)}{\int \frac{1}{\sqrt{(2\pi)^k |(I+M)^{-2}\boldsymbol{\Sigma}|}} \exp\left( -\frac{1}{2}(x - \boldsymbol{\mu})^\top (I + M)^2 \boldsymbol{\Sigma}^{-1}(x - \boldsymbol{\mu}) \right)}$$

$$= \frac{|I + M|^{-1} \exp\left( -\frac{1}{2} x^\top (I + M)^2 \boldsymbol{\Sigma}^{-1} x + \boldsymbol{\mu}^\top (I + M)^2 \boldsymbol{\Sigma}^{-1} x \right)}{\int \exp\left( -\frac{1}{2} x^\top (I + M)^2 \boldsymbol{\Sigma}^{-1} x + \boldsymbol{\mu}^\top (I + M)^2 \boldsymbol{\Sigma}^{-1} x \right)}$$

In terms of the aforementioned $\boldsymbol{T}$ and $\boldsymbol{m}$, and for convenience defining $A = (I + M)^2$:

$$\ell(x) = |A|^{-1/2} + \left( \frac{1}{2} x^\top A \boldsymbol{T} x - \boldsymbol{m}^\top A x \right) - \log\left( \int \exp\left( \frac{1}{2} x^\top A \boldsymbol{T} x - \boldsymbol{m}^\top A x \right) \right)$$

$$\nabla \ell(x) = \begin{bmatrix} \frac{1}{2}(Axx^\top)_{\diagdown} \\ -Ax \end{bmatrix} - \frac{\int \begin{bmatrix} \frac{1}{2}(Axx^\top)_{\diagdown} \\ -Ax \end{bmatrix} \exp\left( \frac{1}{2} x^\top A \boldsymbol{T} x - \boldsymbol{m}^\top A x \right)}{\int \exp\left( \frac{1}{2} x^\top A \boldsymbol{T} x - \boldsymbol{m}^\top A x \right)}$$

$$= \begin{bmatrix} \frac{1}{2}(Axx^\top)_{\diagdown} \\ -Ax \end{bmatrix} - \mathbb{E}_{z \sim \mathcal{N}(\boldsymbol{T}^{-1}\boldsymbol{m}, (\boldsymbol{A}\boldsymbol{T})^{-1})} \begin{bmatrix} \frac{1}{2}(Azz^\top)_{\diagdown} \\ -Az \end{bmatrix}. \tag{21}$$

From here, following an identical argument to [Das+19] Equation (3.7), we find that

$$\boldsymbol{H}_\ell = \mathrm{Cov}_{z \sim \mathcal{N}(\boldsymbol{T}^{-1}\boldsymbol{m}, (AT)^{-1})} \left[ \begin{pmatrix} (-\frac{1}{2}Azz^T)_{\diagdown} \\ z \end{pmatrix}, \begin{pmatrix} (-\frac{1}{2}Azz^T)_{\diagdown} \\ z \end{pmatrix} \right] \succcurlyeq \boldsymbol{0},$$

i.e. that the log-likelihood is indeed convex with respect to $\begin{bmatrix} \boldsymbol{T} \\ \boldsymbol{m} \end{bmatrix}$, as desired. □

### E.3.5 Applying Danskin's Theorem

The previous two parts show that we can indeed apply Danskin's theorem to the outer minimization, and in particular that the gradient of $f$ at $M = M^*$ is in the subdifferential of the outer minimization problem. We proceed by writing out this gradient explicitly, and then setting it to zero (note that since we have shown $f$ is convex for all choices of perturbation, we can use the fact that a convex function is globally minimized $\iff$ its subgradient contains zero). We continue from above,

plugging in (20) for $M$ and using (21) to write the gradients of $\ell$ with respect to $\boldsymbol{T}$ and $\boldsymbol{m}$.

$$0 = \nabla_{\begin{bmatrix}\boldsymbol{T}\\\boldsymbol{m}\end{bmatrix}}\ell = \mathbb{E}_{x\sim\mathcal{N}(\boldsymbol{\mu}^*,\boldsymbol{\Sigma}^*)}\left[\begin{bmatrix}\frac{1}{2}(Axx^\top)_{\diagdown}\\-Ax\end{bmatrix}\right] - \mathbb{E}_{z\sim\mathcal{N}(\boldsymbol{T}^{-1}\boldsymbol{m},(A\boldsymbol{T})^{-1})}\left[\begin{bmatrix}\frac{1}{2}(Azz^\top)_{\diagdown}\\-Az\end{bmatrix}\right]$$

$$= \mathbb{E}_{x\sim\mathcal{N}(\boldsymbol{\mu}^*,\boldsymbol{\Sigma}^*)}\begin{bmatrix}\frac{1}{2}(Axx^\top)_{\diagdown}\\-Ax\end{bmatrix} - \mathbb{E}_{z\sim\mathcal{N}(\boldsymbol{T}^{-1}\boldsymbol{m},(A\boldsymbol{T})^{-1})}\begin{bmatrix}\frac{1}{2}(Azz^\top)_{\diagdown}\\-Az\end{bmatrix}$$

$$= \begin{bmatrix}\frac{1}{2}(A\boldsymbol{\Sigma}_*)_{\diagdown}\\-A\boldsymbol{\mu}_*\end{bmatrix} - \mathbb{E}_{z\sim\mathcal{N}(\boldsymbol{T}^{-1}\boldsymbol{m},(A\boldsymbol{T})^{-1})}\begin{bmatrix}\frac{1}{2}(A(A\boldsymbol{T})^{-1})_{\diagdown}\\-A\boldsymbol{T}^{-1}\boldsymbol{m}\end{bmatrix}$$

$$= \begin{bmatrix}\frac{1}{2}A\boldsymbol{\Sigma}_*\\-A\boldsymbol{\mu}_*\end{bmatrix} - \begin{bmatrix}\frac{1}{2}A(A\boldsymbol{T})^{-1}\\-A\boldsymbol{T}^{-1}\boldsymbol{m}\end{bmatrix}$$

$$= \begin{bmatrix}\frac{1}{2}A\boldsymbol{\Sigma}_* - \frac{1}{2}\boldsymbol{T}^{-1}\\A\boldsymbol{T}^{-1}\boldsymbol{m} - A\boldsymbol{\mu}_*\end{bmatrix} \tag{22}$$

Using this fact, we derive an *implicit* expression for the robust covariance matrix $\boldsymbol{\Sigma}$. Note that for the sake of brevity, we now use $M$ to denote the optimal adversarial perturbation (previously defined as $M^*$ in (20)). This implicit formulation forms the foundation of the bounds given by our main results.

**Lemma 4.** *The minimax problem discussed throughout this work admits the following (implicit) form of solution:*

$$\boldsymbol{\Sigma} = \frac{1}{\lambda}I + \frac{1}{2}\boldsymbol{\Sigma}_* + \sqrt{\frac{1}{\lambda}\boldsymbol{\Sigma}_* + \frac{1}{4}\boldsymbol{\Sigma}_*^2},$$

*where $\lambda$ is such that $M \in \mathcal{M}$, and is thus dependent on $\boldsymbol{\Sigma}$.*

*Proof.* Rewriting (22) in the standard parameterization (with respect to $\boldsymbol{\mu}, \boldsymbol{\Sigma}$) and re-expanding $A = (I + M)^2$ yields:

$$0 = \nabla_{\begin{bmatrix}\boldsymbol{T}\\\boldsymbol{m}\end{bmatrix}}\ell = \begin{bmatrix}\frac{1}{2}(I + M)^2\boldsymbol{\Sigma}_* - \frac{1}{2}\boldsymbol{\Sigma}\\(I + M)^2\boldsymbol{\mu} - (I + M)^2\boldsymbol{\mu}_*\end{bmatrix}$$

Now, note that the equations involving $\boldsymbol{\mu}$ and $\boldsymbol{\Sigma}$ are completely independent, and thus can be solved separately. In terms of $\boldsymbol{\mu}$, the relevant system of equations is $A\boldsymbol{\mu} - A\boldsymbol{\mu}_* = 0$, where multiplying by the inverse $A$ gives that

$$\boldsymbol{\mu} = \boldsymbol{\mu}_*. \tag{23}$$

This tells us that the mean learned via $\ell_2$-robust maximum likelihood estimation is precisely the true mean of the distribution.

Now, in the same way, we set out to find $\boldsymbol{\Sigma}$ by solving the relevant system of equations:

$$\boldsymbol{\Sigma}_*^{-1} = \boldsymbol{\Sigma}^{-1}(M + I)^2. \tag{24}$$

Now, we make use of the Woodbury Matrix Identity in order to write $(I + M)$ as

$$I + (\lambda\boldsymbol{\Sigma} - I)^{-1} = I + \left(-I - \left(\frac{1}{\lambda}\boldsymbol{\Sigma}^{-1} - I\right)^{-1}\right) = -\left(\frac{1}{\lambda}\boldsymbol{\Sigma}^{-1} - I\right)^{-1}.$$

Thus, we can revisit (24) as follows:

$$\boldsymbol{\Sigma}_*^{-1} = \boldsymbol{\Sigma}^{-1}\left(\frac{1}{\lambda}\boldsymbol{\Sigma}^{-1} - I\right)^{-2}$$

$$\frac{1}{\lambda^2}\boldsymbol{\Sigma}_*^{-1}\boldsymbol{\Sigma}^{-2} - \left(\frac{2}{\lambda}\boldsymbol{\Sigma}_*^{-1} + I\right)\boldsymbol{\Sigma}^{-1} + \boldsymbol{\Sigma}_*^{-1} = 0$$

$$\frac{1}{\lambda^2}\boldsymbol{\Sigma}_*^{-1} - \left(\frac{2}{\lambda}\boldsymbol{\Sigma}_*^{-1} + I\right)\boldsymbol{\Sigma} + \boldsymbol{\Sigma}_*^{-1}\boldsymbol{\Sigma}^2 = 0$$

We now apply the quadratic formula to get an implicit expression for $\mathbf{\Sigma}$ (implicit since technically $\lambda$ depends on $\mathbf{\Sigma}$):

$$\mathbf{\Sigma} = \left( \frac{2}{\lambda} \mathbf{\Sigma}_*^{-1} + I \pm \sqrt{\frac{4}{\lambda} \mathbf{\Sigma}_*^{-1} + I} \right) \frac{1}{2} \mathbf{\Sigma}_*$$

$$= \frac{1}{\lambda} I + \frac{1}{2} \mathbf{\Sigma}_* + \sqrt{\frac{1}{\lambda} \mathbf{\Sigma}_* + \frac{1}{4} \mathbf{\Sigma}_*^2}. \tag{25}$$

This concludes the proof. $\qquad\square$

### E.3.6 Bounding $\lambda$

We now attempt to characterize the shape of $\lambda$ as a function of $\varepsilon$. First, we use the fact that $\mathbb{E}[\|Xv\|^2] = \mathrm{tr}(X^2)$ for standard normally-drawn $v$. Thus, $\lambda$ is set such that $\mathrm{tr}(\mathbf{\Sigma}_* M^2) = \varepsilon$, i.e:

$$\sum_{i=0} \frac{\mathbf{\Sigma}_{ii}^*}{(\lambda \mathbf{\Sigma}_{ii} - 1)^2} = \varepsilon \tag{26}$$

Now, consider $\varepsilon^2$ as a function of $\lambda$. Observe that for $\lambda \geq \frac{1}{\sigma_{min}(\mathbf{\Sigma})}$, we have that $M$ must be positive semi-definite, and thus $\varepsilon^2$ decays smoothly from $\infty$ (at $\lambda = \frac{1}{\sigma_{min}}$) to zero (at $\lambda = \infty$). Similarly, for $\lambda \leq \frac{1}{\sigma_{max}(\mathbf{\Sigma})}$, $\varepsilon$ decays smoothly as $\lambda$ *decreases*. Note, however, that such values of $\lambda$ would necessarily make $M$ *negative semi-definite*, which would actually *help* the log-likelihood. Thus, we can exclude this case; in particular, for the remainder of the proofs, we can assume $\lambda \geq \frac{1}{\sigma_{max}(\mathbf{\Sigma})}$.

Also observe that the zeros of $\varepsilon$ in terms of $\lambda$ are only at $\lambda = \pm\infty$. Using this, we can show that there exists some $\varepsilon_0$ for which, for all $\varepsilon < \varepsilon_0$, the only corresponding possible valid value of $\lambda$ is where $\lambda \geq \frac{1}{\sigma_{min}}$. This idea is formalized in the following Lemma.

**Lemma 5.** *For every $\mathbf{\Sigma}_*$, there exists some $\varepsilon_0 > 0$ for which, for all $\varepsilon \in [0, \varepsilon_0)$ the only admissible value of $\lambda$ is such that $\lambda \geq \frac{1}{\sigma_{min}(\mathbf{\Sigma})}$, and thus such that $M$ is positive semi-definite.*

*Proof.* We prove the existence of such an $\varepsilon_0$ by lower bounding $\varepsilon$ (in terms of $\lambda$) for any finite $\lambda > 0$ that does not make $M$ PSD. Providing such a lower bound shows that for small enough $\varepsilon$ (in particular, less than this lower bound), the only corresponding values of $\lambda$ are as desired in the statement[8].

In particular, if $M$ is not PSD, then there must exist at least one index $k$ such that $\lambda \mathbf{\Sigma}_{kk} < 1$, and thus $(\lambda \mathbf{\Sigma}_{kk} - 1)^2 \leq 1$ for all $\lambda > 0$. We can thus lower bound (26) as:

$$\varepsilon = \sum_{i=0} \frac{\mathbf{\Sigma}_{ii}^*}{(\lambda \mathbf{\Sigma}_{ii} - 1)^2} \geq \frac{\mathbf{\Sigma}_{kk}^*}{(\lambda \mathbf{\Sigma}_{kk} - 1)^2} \geq \mathbf{\Sigma}_{kk}^* \geq \sigma_{min}(\mathbf{\Sigma}^*) > 0 \tag{27}$$

By contradiction, it follows that for any $\varepsilon < \sigma_{min}(\mathbf{\Sigma}_*)^2$, the only admissible $\lambda$ is such that $M$ is PSD, i.e. according to the statement of the Lemma. $\qquad\square$

In the regime $\varepsilon \in [0, \varepsilon_0)$, note that $\lambda$ is inversely proportional to $\varepsilon$ (i.e. as $\varepsilon$ grows, $\lambda$ decreases). This allows us to get a qualitative view of (25): as the allowed perturbation value increases, the robust covariance $\mathbf{\Sigma}$ resembles the identity matrix more and more, and thus assigns more and more variance on initially low-variance features. The $\sqrt{\mathbf{\Sigma}_*}$ term indicates that the robust model also adds uncertainty proportional to the square root of the initial variance—thus, low-variance features will have (relatively) more uncertainty in the robust case. Indeed, our main result actually follows as a (somewhat loose) formalization of this intuition.

### E.3.7  Proof of main theorems

First, we give a proof of Theorem 2, providing lower and upper bounds on the learned robust covariance $\mathbf{\Sigma}$ in the regime $\varepsilon \in [0, \varepsilon_0)$.

**Theorem 2** (Robustly Learned Parameters). *Just as in the non-robust case, $\boldsymbol{\mu}_r = \boldsymbol{\mu}^*$, i.e. the true mean is learned. For the robust covariance $\mathbf{\Sigma}_r$, there exists an $\varepsilon_0 > 0$, such that for any $\varepsilon \in [0, \varepsilon_0)$,*

$$\mathbf{\Sigma}_r = \frac{1}{2}\mathbf{\Sigma}_* + \frac{1}{\lambda}\cdot \boldsymbol{I} + \sqrt{\frac{1}{\lambda}\cdot \mathbf{\Sigma}_* + \frac{1}{4}\mathbf{\Sigma}_*^2}, \qquad where \qquad \Omega\left(\frac{1+\varepsilon^{1/2}}{\varepsilon^{1/2}+\varepsilon^{3/2}}\right) \le \lambda \le O\left(\frac{1+\varepsilon^{1/2}}{\varepsilon^{1/2}}\right).$$

*Proof.* We have already shown that $\boldsymbol{\mu} = \boldsymbol{\mu}_*$ in the robust case (c.f. (23)). We choose $\varepsilon_0$ to be as described, i.e. the largest $\varepsilon$ for which the set $\{\lambda : \text{tr}(\mathbf{\Sigma}_*^2 M) = \varepsilon, \lambda \ge 1/\sigma_{\max}(\mathbf{\Sigma})\}$ has only one element $\lambda$ (which, as we argued, must not be less than $1/\sigma_{min}(\mathbf{\Sigma})$). We have argued that such an $\varepsilon_0$ must exist.

We prove the result by combining our early derivation (in particular, (24) and (25)) with upper and lower bound on $\lambda$, which we can compute based on properties of the trace operator. We begin by deriving a lower bound on $\lambda$. By linear algebraic manipulation (given in Appendix E.3.8), we get the following bound:

$$\lambda \ge \frac{d}{\text{tr}(\mathbf{\Sigma})}\left(1 + \sqrt{\frac{d\cdot\sigma_{min}(\mathbf{\Sigma}_*)}{\varepsilon}}\right) \tag{28}$$

Now, we can use (24) in order to remove the dependency of $\lambda$ on $\mathbf{\Sigma}$:

$$\mathbf{\Sigma} = \mathbf{\Sigma}_*(M + I)^2$$
$$\text{tr}(\mathbf{\Sigma}) = \text{tr}\left[(\mathbf{\Sigma}_*^{1/2}M + \mathbf{\Sigma}_*^{1/2})^2\right]$$
$$\le 2\cdot\text{tr}\left[(\mathbf{\Sigma}_*^{1/2}M)^2 + (\mathbf{\Sigma}_*^{1/2})^2\right]$$
$$\le 2\cdot(\varepsilon + \text{tr}(\mathbf{\Sigma}_*)).$$

Applying this to (28) yields:

$$\lambda \ge \frac{d/2}{\varepsilon + \text{tr}(\mathbf{\Sigma}_*)}\left(1 + \sqrt{\frac{d\cdot\sigma_{min}(\mathbf{\Sigma}_*)}{\varepsilon}}\right).$$

Note that we can simplify this bound significantly by writing $\varepsilon = d\cdot\sigma_{min}(\mathbf{\Sigma}_*)\varepsilon' \le \text{tr}(\mathbf{\Sigma}_*)\varepsilon'$, which does not affect the result (beyond rescaling the valid regime $(0, \varepsilon_0)$), and gives:

$$\lambda \ge \frac{d/2}{(1+\varepsilon')\text{tr}(\mathbf{\Sigma}_*)}\left(1 + \frac{1}{\sqrt{\varepsilon'}}\right) \ge \frac{d\cdot(1+\sqrt{\varepsilon'})}{2\sqrt{\varepsilon'}(1+\varepsilon')\text{tr}(\mathbf{\Sigma}_*)}$$

Next, we follow a similar methodology (Appendix E.3.8) in order to upper bound $\lambda$:

$$\lambda \le \frac{1}{\sigma_{min}(\mathbf{\Sigma})}\left(\sqrt{\frac{\|\mathbf{\Sigma}_*\|_F\cdot d}{\varepsilon}} + 1\right).$$

Note that by (24) and positive semi-definiteness of $M$, it must be that $\sigma_{min}(\mathbf{\Sigma}) \ge \sigma_{min}(\mathbf{\Sigma}_*)$. Thus, we can simplify the previous expression, also substituting $\varepsilon = d\cdot\sigma_{min}(\mathbf{\Sigma}_*)\varepsilon'$:

$$\lambda \le \frac{1}{\sigma_{min}(\mathbf{\Sigma}_*)}\left(\sqrt{\frac{\|\mathbf{\Sigma}_*\|_F}{\sigma_{min}(\mathbf{\Sigma}_*)\varepsilon'}} + 1\right) = \frac{\|\mathbf{\Sigma}_*\|_F + \sqrt{\varepsilon\cdot\sigma_{min}(\mathbf{\Sigma}_*)}}{\sigma_{min}(\mathbf{\Sigma}_*)^{3/2}\sqrt{\varepsilon}}$$

These bounds can be straightforwardly combined with Lemma 4, which concludes the proof. $\square$

Using this theorem, we can now show Theorem 3:

**Theorem 3** (Gradient alignment). *Let $f(x)$ and $f_r(x)$ be monotonic classifiers based on the linear separator induced by standard and $\ell_2$-robust maximum likelihood classification, respectively. The maximum angle formed between the gradient of the classifier (wrt input) and the vector connecting the classes can be smaller for the robust model:*

$$\min_{\boldsymbol{\mu}}\frac{\langle\boldsymbol{\mu}, \nabla_x f_r(x)\rangle}{\|\boldsymbol{\mu}\|\cdot\|\nabla_x f_r(x)\|} > \min_{\boldsymbol{\mu}}\frac{\langle\boldsymbol{\mu}, \nabla_x f(x)\rangle}{\|\boldsymbol{\mu}\|\cdot\|\nabla_x f(x)\|}.$$

*Proof.* To prove this, we make use of the following Lemmas:

**Lemma 6.** *For two positive definite matrices $A$ and $B$ with $\kappa(A) > \kappa(B)$, we have that $\kappa(A+B) \leq \max\{\kappa(A), \kappa(B)\}$.*

*Proof.* We proceed by contradiction:

$$\kappa(A + B) = \frac{\lambda_{max}(A) + \lambda_{max}(B)}{\lambda_{min}(A) + \lambda_{min}(B)}$$

$$\kappa(A) = \frac{\lambda_{max}(A)}{\lambda_{min}(A)}$$

$$\kappa(A) \geq \kappa(A + B)$$

$$\iff \lambda_{max}(A)\left(\lambda_{min}(A) + \lambda_{min}(B)\right) \geq \lambda_{min}(A)\left(\lambda_{max}(A) + \lambda_{max}(B)\right)$$

$$\iff \lambda_{max}(A)\lambda_{min}(B) \geq \lambda_{min}(A)\lambda_{max}(B)$$

$$\iff \frac{\lambda_{max}(A)}{\lambda_{min}(A)} \geq \frac{\lambda_{min}(A)}{\lambda_{max}(B)},$$

which is false by assumption. This concludes the proof. $\qquad\square$

**Lemma 7** (Straightforward). *For a positive definite matrix $A$ and $k > 0$, we have that*

$$\kappa(A + k \cdot I) < \kappa(A) \qquad \kappa(A + k \cdot \sqrt{A}) \leq \kappa(A).$$

**Lemma 8** (Angle induced by positive definite matrix; folklore). [9] *For a positive definite matrix $A \succ 0$ with condition number $\kappa$, we have that*

$$\min_{x} \frac{x^\top A x}{\|Ax\|_2 \cdot \|x\|_2} = \frac{2\sqrt{\kappa}}{1 + \kappa}. \tag{29}$$

These two results can be combined to prove the theorem. First, we show that $\kappa(\mathbf{\Sigma}) \leq \kappa(\mathbf{\Sigma}_*)$:

$$\kappa(\mathbf{\Sigma}) = \kappa\left(\frac{1}{\lambda}I + \frac{1}{2}\mathbf{\Sigma}_* + \sqrt{\frac{1}{\lambda}\mathbf{\Sigma}_* + \frac{1}{4}\mathbf{\Sigma}_*^2}\right)$$

$$< \max\left\{\kappa\left(\frac{1}{\lambda}I + \frac{1}{2}\mathbf{\Sigma}_*\right), \kappa\left(\sqrt{\frac{1}{\lambda}\mathbf{\Sigma}_* + \frac{1}{4}\mathbf{\Sigma}_*^2}\right)\right\}$$

$$< \max\left\{\kappa\left(\mathbf{\Sigma}_*\right), \sqrt{\kappa\left(\frac{1}{\lambda}\mathbf{\Sigma}_* + \frac{1}{4}\mathbf{\Sigma}_*^2\right)}\right\}$$

$$= \max\left\{\kappa\left(\mathbf{\Sigma}_*\right), \sqrt{\kappa\left(\frac{2}{\lambda}\sqrt{\frac{1}{4}\mathbf{\Sigma}_*^2} + \frac{1}{4}\mathbf{\Sigma}_*^2\right)}\right\}$$

$$\leq \kappa\left(\mathbf{\Sigma}_*\right).$$

Finally, note that (29) is a strictly decreasing function in $\kappa$, and as such, we have shown the theorem.
$\qquad\square$

### E.3.8 Bounds for $\lambda$

**Lower bound.**

$$
\begin{aligned}
\varepsilon &= \text{tr}(\mathbf{\Sigma}_* M^2) & \\
&\geq \sigma_{min}(\mathbf{\Sigma}_*) \cdot \text{tr}(M^2) & \text{by the definition of tr}(\cdot) \\
&\geq \frac{\sigma_{min}(\mathbf{\Sigma}_*)}{d} \cdot \text{tr}(M)^2 & \text{by Cauchy-Schwarz} \\
&\geq \frac{\sigma_{min}(\mathbf{\Sigma}_*)}{d} \cdot \left[ \text{tr}\left( (\lambda\mathbf{\Sigma} - \boldsymbol{I})^{-1} \right) \right]^2 & \text{Expanding } M \text{ (20)} \\
&\geq \frac{\sigma_{min}(\mathbf{\Sigma}_*)}{d} \cdot \left[ \text{tr}\left( \lambda\mathbf{\Sigma} - \boldsymbol{I} \right)^{-1} \cdot d^2 \right]^2 & \text{AM-HM inequality} \\
&\geq d^3 \cdot \sigma_{min}(\mathbf{\Sigma}_*) \cdot \left[ \lambda \cdot \text{tr}(\mathbf{\Sigma}) - d \right]^{-2} & \\
\left[ \lambda \cdot \text{tr}(\mathbf{\Sigma}) - d \right]^2 &\geq \frac{d^3 \cdot \sigma_{min}(\mathbf{\Sigma}_*)}{\varepsilon} & \\
\lambda \cdot \text{tr}(\mathbf{\Sigma}) - d &\geq \frac{d^{3/2} \cdot \sqrt{\sigma_{min}(\mathbf{\Sigma}_*)}}{\sqrt{\varepsilon}} & \text{since } M \text{ is PSD} \\
\lambda &\geq \frac{d}{\text{tr}(\mathbf{\Sigma})} \left( 1 + \sqrt{\frac{d \cdot \sigma_{min}(\mathbf{\Sigma}_*)}{\varepsilon}} \right) &
\end{aligned}
$$

**Upper bound**

$$
\begin{aligned}
\varepsilon &= \text{tr}(\mathbf{\Sigma}_* M^2) \\
&\leq \|\mathbf{\Sigma}_*\|_F \cdot d \cdot \sigma_{max}(M)^2 \\
&\leq \|\mathbf{\Sigma}_*\|_F \cdot d \cdot \sigma_{min}(M)^{-2} \\
\lambda \cdot \sigma_{min}(\mathbf{\Sigma}) - 1 &\leq \sqrt{\frac{\|\mathbf{\Sigma}_*\|_F \cdot d}{\varepsilon}} \\
\lambda &\leq \frac{1}{\sigma_{min}(\mathbf{\Sigma})} \left( \sqrt{\frac{\|\mathbf{\Sigma}_*\|_F \cdot d}{\varepsilon}} + 1 \right).
\end{aligned}
$$

## Footnotes

[7]We use 10k steps to construct the dataset from noise, instead to using 1k steps done when the input is a different training set image (cf. Table 5).

[8]Since our only goal is existence, we lose many factors from the analysis that would give a tighter bound on $\varepsilon_0$.

[9]A proof can be found in `https://bit.ly/2L6jdAT`