[Reviews · NeurIPS 2019]

Reviewer 1



Over the past few years, adversarial examples have received a significant amount of attention in the deep learning community. This paper approaches and addresses this important problem in a unique way by disentangling robust and non-robust features in a standard dataset. I have few queries: By selecting robust or non-robust features for standard or adversarial training, how do you avoid over-fitting of models to these features? In your proposed method, you randomly sample clean images from the distribution as the starting point of optimisation. Do the obtained images look similar to the source images or target images (images which provide robust features in the optimization)? If they are similar to the source images, dosen't it mean that the robust features are not robust? Why do the authors use distance in robust feature space as your optimisation objective? Any specific reason or motivation for this? UPDATE AFTER REBUTTAL: I thank the authors for addressing my comments. They have original contribution, and overall research problem has been nicely presented and addressed. I am increasing my score by 1 for this paper.

Reviewer 2



Quality : The paper is technically sound and tackles a non-trivial and important issue Significance : good (see comment on the contributions) Originality : The contributions are original (but it would be better to position the paper in the main body, not the in the appendices) Clarity : While the English is good and sentences are in general pleasant to read, the paper lacks in clarity mainly because it is poorly organized. Important aspects of the papers can only be found in the appendices making the paper main body difficult to fully understand and not self-contained. The most important aspects in question are : - review of prior art allowing to position the paper - Algorithms allowing to generate modified inputs for sections 3.1 and 3.2 Clarity would also be greatly improved by exemplifying things. Experiments are made in a deep learning context. It would be good to explain the introduced concepts in this setting progressively throughout the paper. Comment : utility and robustness analysis are made feature by feature. The most discriminative information is often obtained by combining features. For example, take a two concentric circles dataset (such as those generated by the make_circles function from sklearn). Each raw feature achieves zero usefulness. The feature corresponding to the distance from the circles origin achieves maximal utility and is obtained by mixing raw features. If the function space in which features live is extremely general, it can be argued that any combination of the raw features belongs to it and therefore the definition matches its purpose. BTW, I believe that a feature with maximal utility is any non-rescaled version of p(y|x). I don’t think that it challenges the validity of the authors’ contributions but maybe they should comment on that because the definition might, at first sight, seem too simplistic. Remarks : Following line 83, investigated classifiers are perceptron like. To what extent is this a limit of the authors’ analysis ? Is this general definition useful for other parts of the paper ? Fig. 2 / right side : ordering of methods in the caption and in the figure are not consistent. What is the adversarial technique specifically used to obtain this figure ? I later found out that this is explained in appendix C but is can be briefly mentioned in the caption. Line 84 : it’s unclear to me that the features are learned from the classifier … Parameters w_f are learned but not mappings f. Line 107 : « vulnerability is caused by non-robust features and is not inherently tied to the standard training framework ». Isn’t there training frameworks prone to produce non-robust features ? (Sounds like chicken or the egg causality dilemma) Line 124->135 : very unclear . Some explanations from appendix C need to be brought back here.
 Line 159 : The definition of the deterministically modified class labels t is ambiguous. Do we have t = -y, or something more subtle ? -> again clarified in appendix C Fig. 3 : how is transfer performed. What is transfer success rate ? A short recap would be appreciated (can be in the appendices) The theoretical analysis of section 4 is interesting and illustrative yet based on a particular (and very simple) setting. To what extent can we generalize these results to more general situations ? UPDATE AFTER REBUTTAL: I thank the authors for their additional comments. I am confident that they can improve clarity as they committed to in their feedback and that their work can lead to interesting further developments. score +=1

Reviewer 3



This paper offers a new perspective on the phenomenon of adversarial examples, imperceptible small feature perturbations that cause state-of-the-art models to output incorrect predictions. While previous work related this phenomenon to peculiarities of high-dimensional spaces or the local linearity of the models, this paper posits that adversarial examples are in fact due to sensitivity of models to *well-generalizing* features. In order to present their results, the authors define *robust* features as features that remain useful under adversarial manipulation. The central hypothesis is that there exists both robust and non-robust features for image classifications, and the authors present empirical evidence for this premise by explicitly disentangling both set of features. First, they construct a “robustified” dataset by leveraging a pre-trained robust classifier to explicitly remove non-robust features. More specifically, they construct robust features by optimizing the input (with gradient descent) to match the penultimate layer of the pre-trained robust classifier. Next, they train a standard classifier on these robust features and show that the resulting model achieves strong standard classification and *robust* classification (i.e. is resistant to adversarial examples). Second, the paper introduces a “non-robust” dataset by exploiting a pre-trained classifier to add adversarial perturbations to existing inputs (and relabeling the corresponding outputs, either uniform randomly or deterministically according to the original class label). As a consequence, these examples appear to be mislabeled to humans. They then train a standard classifier on these non-robust features, and show that these models still generalize on the original test set. In other words, even if the robust features are distracting the training signal, these models learn to pick-up non-robust features (which still achieve strong performance). Finally, the paper also studies this phenomenon for maximum likelihood classification for two Gaussian distributions. They show that 1) adversarial vulnerability is essentially explained by the difference between the data-induced metric and the l2 metric and 2) that the gradients for more robust models are better aligned with the adversary’s metric (which support some recent empirical observations). Strengths: 1. A very well-written paper with a new thought-provoking perspective on adversarial examples 2. A very creative and thorough set of experiments to support their claims Questions: 1. Looking at the examples of the robustified dataset, they appear to be more prototypical examples of the classes. Did you observe that the produced examples are less diverse than the original dataset? 2. The classifier trained on robust features appears to be more adversarially robust on ImageNet than of Cifar10 (comparing Fig. 2 with Fig. 12 in supplementary material here). Is there a good explanation for this difference? 3. I’m intrigued by the experiments on the non-robustified dataset, as the robust features now confuse the training signal for the classifier. Although the experiments show that the classifier mostly relies on the non-robust features, I’m wondering how sensitive this result is to the epsilon parameter in the adversarial generation process? UPDATE AFTER REBUTTAL: Thanks for addressing my questions. I'm keeping my score.

[Author Response · NeurIPS 2019]

We want to thank the reviewers for their thorough comments and suggestions for improving the manuscript. We have inlined responses to the major points below, and will address all minor points in our next revision as well.

**(R1) Assumptions** While we cannot establish formal guarantees when constructing our "robust dataset" in Section 3.1 (which we presume this comment is referring to), our method follows a fairly well-motivated approach—-for each input in the original training set, we choose as a seed an image that is randomly selected (independent of label—to avoid introducing any feature-label correlation), and then modify this image to make it match the representation of the original input under our robust model. The resulting dataset thus matches the original in terms of the features used by the robust model, while preventing the re-introduction of features that robust model is invariant to (which are non-robust features).

Finally, it is important to note that, in the end, our result is of *existential* nature: i.e., for the first time, we managed to construct a dataset that results in models that can tackle a non-trivial task and are robust after just *standard* (ERM) training. This suggests that our overarching conceptual framework might be indeed predictive of the way the underlying phenomenon behaves.

**(R1) By selecting ... features?** Note that our definition of a feature defines it via its *generalization* performance. This makes it impossible to "overfit" to a feature in the traditional sense.

**(R1) In your proposed method ... not robust?** This is correct—fortunately, they all resemble the target images.

**(R1) Why distance in robust feature space?** Our goal is to create a training set which does not contain the features that a robust model is invariant to (i.e., non-robust features). Optimizing distance in robust feature space is a clean way to induce this invariance while still matching the features that *are* important for robust classification.

**(R2) Clarity** We want to thank the reviewer for their suggestions regarding clarity of our presentation. In addition to adding examples/exposition around our methods, we will also: make sure to move the related work into the main body (in order to better position our work), and include algorithms for generating the four datasets constructed in the main body of the paper.

**(R2) definition of "feature"** Our formal definition of robust (and non-robust) features in Section 2 is designed to be a high-level guiding framework for the design and analysis of our experiments. As such, there are some nuances/complicated scenarios not captured by our simple definitions (as the reviewer points out), but we viewed it as fully sufficient to describe and predict the results of our experiments. Nonetheless, we view coming up with a more nuanced/fine-grained definition of features as an important direction for future work. As far as our manuscript goes, we will update it to reflect that view (and highlight the corresponding line of future work).

**(R2) Section 4** The goal of S4, as the reviewer points out, is to provide an illustrative example of how misalignment between the "feature metric" in the data and the "adversary metric" (Euclidean distance) can lead to adversarial vulnerability—-and how robust training can "fix" this misalignment. To this end, we settled on the simplest possible setting (e.g. convexity, to ensure a closed-form solution exists even for the robust problem), so that robustness and robust optimization could be studied as rigorously as possible. (It turns out that even in this simple setting analysis is not completely straightforward.) We very much agree with the reviewer that similar analyses for more complicated settings and classifiers would be an important direction for future work.

Nonetheless, our preliminary empirical and theoretical work indicates that the results do extend beyond the simple setting presented here (for example, one can show that for linear models, non-robustness arises from misalignment not only in the case where the data is Gaussian but for any distribution with bounded second moment). While we are happy to include these extensions in the next revision, any more substantial extensions (such as moving beyond linear models, analyzing robust training for different distributions) might warrant separate work.

**(R3) Looking ... original dataset?** While we didn't notice a significant decrease in diversity (the four random samples do look somewhat "prototypical" but this seems to be mostly by chance—we can include a larger selection of random samples in our final version). It's possible that there is a slight decrease in diversity (maybe that's also why *standard* accuracy on the $\mathcal{D}_R$ dataset is very slightly worse than that of the original robust network). It would be interesting to see if different methods of constructing $\mathcal{D}_R$ (for example, starting from a bunch of different random images per training set image, etc.) would be effective at introducing more diversity into the training samples.

**(R3) I'm intrigued ... generation process?** While we did not perform a formal study on this, we noticed that the accuracy increases from 0 (when $\epsilon = 0$, clearly, since at this point it is just training with mislabeled data) and then just plateaus at a reasonable $\epsilon$, a bit higher than the one we used for generating the dataset. (Note that we didn't tune the $\epsilon$ parameter at all to obtain these results, and as a sanity check our results are stable over a reasonable range of $\epsilon$ values.)

[Meta-Review · NeurIPS 2019]

A very good paper, on which all reviewers early agreed on its merits and originality. I strongly suggest the the authors follow on reviews and implement the suggestions they proposed to do, in particular those that allowed score increase (R#2) and, as much as possible, those suggested in L37-L40 of their rebuttal, at least as additional remark in the main file if the space allows it (eventually amending the conclusion to reflect some of the work mentioned). It could represent a great additional appeal for the interested reader and would surely contribute to prepare some feedback and questions for the conference.